# Reinforcement Learning for Better Verbalized Confidence in Long-Form Generation

## Abstract

Hallucination remains a major challenge for the safe and trustworthy deployment of large language models (LLMs) in factual content generation. Prior work has explored confidence estimation as an effective approach to hallucination detection, but often relies on post-hoc self-consistency methods that require computationally expensive sampling. Verbalized confidence offers a more efficient alternative, but existing approaches are largely limited to short-form question answering (QA) tasks and do not generalize well to open-ended generation. In this paper, we propose `LoVeC` (**Lo**ng-form **Ve**rbalized **C**onfidence), an on-the-fly verbalized confidence estimation method for long-form generation. Specifically, we use reinforcement learning (RL) to train LLMs to append numerical confidence scores to each generated statement, serving as a direct and interpretable signal of the factuality of generation. We introduce two novel evaluation settings, *free-form tagging* and *iterative tagging*, to assess different verbalized confidence estimation methods. Experiments on three long-form QA datasets show that our RL-trained models achieve better calibration and generalize robustly across domains. Also, our method is highly efficient, being $20\times$ faster than traditional self-consistency methods while achieving better calibration.

## 1 Introduction

While large language models (LLMs) demonstrate impressive performance across a wide range of tasks (Touvron et al., 2023; Jiang et al., 2023; OpenAI, 2022), one of their most critical limitations is hallucinations (Zhang et al., 2023; Huang et al., 2023). When faced with unfamiliar or uncertain input, LLMs often generate fabricated or incorrect content. These hallucinations pose a significant barrier to the real-world deployment of LLMs (Manakul et al., 2023; Zhang et al., 2024a;b; Yang et al., 2024; 2025), especially in high-stakes domains such as medicine, law, and finance, where factual inaccuracies can have serious consequences (Zhang et al., 2024a;b; Yang et al., 2024).

Reliable confidence and uncertainty estimation is thus crucial for improving the trustworthiness and practical applicability of LLMs. Following the definitions by Lin et al. (2023), uncertainty refers to the variability or dispersion in the model's predictions given *only the input query*. In contrast, confidence is defined with respect to *both the input and the specific generated output*, capturing how certain the model is about that particular response. While much prior research on confidence estimation has focused on short-form question answering (QA) tasks, long-form QA (with outputs exceeding 100 words) is generally more common and better aligned with real-world applications (Zhang et al., 2024a;b; Yang et al., 2024; 2025). However, methods for short-form QA are designed to produce *a single score for an entire response*, and thus **cannot** be naturally generalized to long-form generation with fine-grained confidence estimation.

Recently, there has been growing interest in confidence estimation methods for long-form outputs (Zhang et al., 2024a;b; Jiang et al., 2024; Fadeeva et al., 2024; Liu et al., 2024). A key limitation of existing approaches is that they are often **post-hoc and computationally expensive**. Many rely on generating multiple samples for consistency checking (Zhang et al., 2024a;b; Jiang et al., 2024), or require an additional model (*e.g.*, GPT-4 (OpenAI, 2023)) to extract atomic claims (Fadeeva et al., 2024; Liu et al., 2024). In contrast, verbalized confidence offers a potentially more efficient alternative, as it avoids both multiple sampling and auxiliary models. However, verbalized confidence

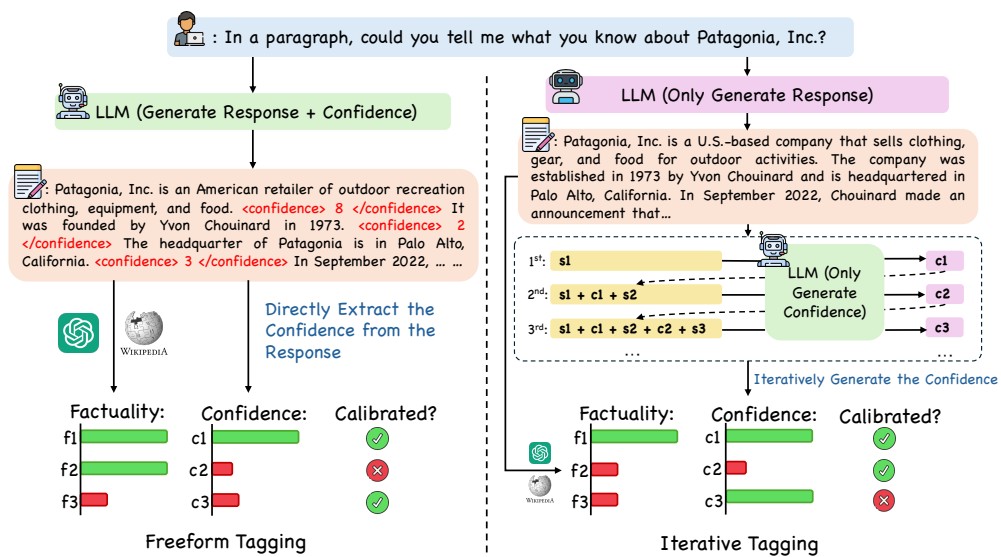

Figure 1: Overview of our two evaluation settings. In *Free-form Tagging*, the model generates both the answer and confidence score suffix. In *Iterative Tagging*, the model is given a fixed response and assigns confidence scores sentence-by-sentence.

remains underexplored in the context of long-form text generation, and it is unclear whether it can provide well-calibrated confidence estimates.

To address these challenges, we propose `LoVeC` (**Lo**ng-form **Ve**rbalized **C**onfidence), an *on-the-fly* verbalized confidence estimation method that generates confidence scores alongside long-form factual statements in a single decoding pass (Contribution **#1**). Specifically, we apply a reinforcement learning (RL)-based approach that enables LLMs to produce well-calibrated confidence estimates during text generation (Figure 1). Compared to supervised fine-tuning (SFT), RL enables direct optimization toward task-specific reward signals, aligning model behavior with desired outcomes beyond token-level likelihoods (Rafailov et al., 2023b; Cao et al., 2024). Moreover, RL does not require fine-grained token-level annotations, which are often expensive or unavailable in practice (Lee et al., 2023; Kirk et al., 2023). We design both off-policy (DPO) and on-policy (GRPO) RL training strategies to accommodate scenarios with or without an oracle fact-checker.

Another key challenge in confidence calibration lies in the fair and rigorous evaluation of different models and methods. To this end, we propose two novel evaluation settings for verbalized confidence estimation in long-form generation (Contribution **#2**; illustrated in Figure 1): *free-form tagging* and *iterative tagging*. In free-form tagging, the model is prompted with a question and generates a complete answer with verbalized confidence tags. Since different models may produce different outputs under this setting, direct comparison can be challenging. To address scenarios where a fixed long-form response is required, we introduce *iterative tagging*, a novel setting in which the model is provided with a fixed answer and tasked with assigning confidence scores sentence-by-sentence.

Our experiments (§5) on Llama-3-8B-Instruct (Meta, 2024) and Gemma-2-9B-It (Team et al., 2024), evaluated across three in-domain and out-of-domain long-form QA datasets, demonstrate better calibration in both iterative and free-form tagging. Our analysis further shows that `LoVeC` is highly efficient, achieving a $20\times$ speedup compared to state-of-the-art methods, and generalizes well to short-form QA tasks. In our analysis (§6), we also investigate why RL outperforms SFT in our case and provide practical insights for future applications.

## 2 RELATED WORK

**Confidence/Uncertainty Estimation in Long-form Generations.** Previous research on confidence and uncertainty estimation has primarily focused on multiple-choice or short-form question answering (Lin et al., 2023; Murray & Chiang, 2018; Kuhn et al., 2023; Vazhentsev et al., 2023; Duan et al.,

2023; Zhu et al., 2023; Xiong et al., 2024; Tian et al., 2023; Ulmer et al., 2024). Recently, there has been increasing interest in confidence and uncertainty estimation for long-form generation. Zhang et al. (2024a) propose LUQ, an uncertainty estimation method designed for long-form generation at both the sentence and passage levels. This approach requires sampling multiple responses, making it computationally expensive. Several studies (Zhang et al., 2024b; Jiang et al., 2024; Fadeeva et al., 2024; Liu et al., 2024) explore post-hoc methods that estimate claim-level uncertainty in long-form outputs. While these approaches offer finer-grained confidence estimates, they typically rely on GPT-based claim extraction, leading to high computational cost. In contrast, we propose an *on-the-fly* verbalized confidence estimation method that generates confidence scores alongside long-form factual statements *in a single decoding pass*. Our methods do not need additional sampling or API calling, making it more efficient and scalable.

**Verbalized Confidence Estimation.** Teaching LLMs to verbalize their confidence has been widely explored in short-form generation (Xiong et al., 2024; Tian et al., 2023; Cheng et al., 2024; Chen et al., 2024; Li et al., 2024; Lin et al., 2022; Xu et al., 2024; Zhang et al., 2024c; Han et al., 2024; Stangel et al., 2025). However, *extending verbalized uncertainty to long-form generation remains challenging*, as multiple aspects may vary in certainty within a single response. Recent work addresses this problem by tightly coupling uncertainty cues with the generated output. LoGU (Yang et al., 2024) trains models to flag uncertain claims during generation, and Band et al. (2024) propose linguistic calibration by embedding expressions such as "I believe" or "I am 70% uncertain" into the text. Although both approaches improve human interpretability, they lack machine interpretability, making post-processing and integration with downstream tasks more difficult. In contrast, our method produces structured outputs by appending numerical confidence tags to each sentence, offering greater flexibility and interpretability.

**Reinforcement Learning for Confidence Estimation** Reinforcement learning (RL) is increasingly used to fine-tune LLMs, often outperforming supervised fine-tuning (SFT) when target behaviors can be sampled from the base model (Cao et al., 2024; Ouyang et al., 2022; Setlur et al., 2025; Guo et al., 2025). Confidence estimation via RL is still new and mostly studied in short-form QA. PPO-based methods such as RewardingDoubt (Stangel et al., 2025) and SaySelf (Xu et al., 2024) outperform SFT techniques like R-tuning (Zhang et al., 2024d), but work on long-form confidence remains limited. LoGU (Yang et al., 2024) applies direct preference optimization (DPO) (Rafailov et al., 2023b) to generate ordinal phrases, while Band et al. (2024) use PPO to calibrate user-facing answers. However, these approaches rely on text-embedded outputs that are difficult to process and evaluate systematically. In contrast, we use DPO and group relative policy optimization (GRPO) (Shao et al., 2024) to append a bounded numerical confidence score after each statement.

## 3 PRELIMINARIES

In this section, we introduce the preliminaries of confidence estimation in long-form generation.

**Primary Goal.** In long-form confidence estimation, the primary objective is to align confidence scores with the factuality of the generated output (Zhang et al., 2024a;b; Yang et al., 2024; Huang et al., 2024b; Jiang et al., 2024; Fadeeva et al., 2024; Liu et al., 2024). The focus on factuality is mainly for two reasons: (1) hallucinations remain a significant challenge in LLMs, and confidence estimation can effectively indicate potential hallucinations during generation; (2) the factuality of a sentence can be objectively assessed, enabling a more quantitative and consistent evaluation compared to subjective criteria such as creativity or coherence (Zhang et al., 2024b; Yang et al., 2024).

**Granularity.** Formally, given an input query $q$, an LLM parameterized by $\theta$ generates a response $y = \pi_\theta(q)$. Confidence estimation can be performed at various granularities depending on whether the confidence score is assigned at the level of atomic claims (a short sentence conveying a single piece of information) (Zhang et al., 2024b; Jiang et al., 2024; Fadeeva et al., 2024), for each sentence (Zhang et al., 2024a; Manakul et al., 2023), or the whole passage (Zhang et al., 2024a; Huang et al., 2024b). For sentence-level confidence estimation, the response $y$ is defined as: $y = \pi_\theta(q)$ consisting of a sequence of sentences $\mathbf{s}$ and corresponding confidence scores $\mathbf{c}$:

$$y = \{(s_1, c_1), (s_2, c_2), \ldots, (s_n, c_n)\} = \{(s_i, c_i)\}_{i=1}^n. \tag{1}$$

where $s_i$ represents the $i^{th}$ sentence and $c_i \in [0, 1]$ denotes the corresponding confidence score, representing the estimated probability of factual correctness; higher values indicate greater confidence.

**Factuality Evaluation.** Each sentence $s_i$ is assigned a factuality score $f_i \in [0, 1]$, reflecting its actual factual accuracy. These factual scores $\mathbf{f}$ are obtained by prompting an oracle verification model $\mathcal{O}$ with suitable supporting evidence $E$ pertinent to the query $q$:

$$\mathbf{f} = \texttt{FactCheck}(\mathcal{O}, q, E, \mathbf{s}). \tag{2}$$

**Confidence Evaluation.** To evaluate these confidence scores, the objective is to ensure the confidence scores $c_i$ generated by the model are well-calibrated and closely align with the independently determined factuality scores $f_i$. This calibration requirement is expressed as:

$$\forall i \in \{1, 2, \dots, |\mathbf{f}|\}, c_i \approx f_i, |\mathbf{f}| = |\mathbf{c}| \tag{3}$$

Various metrics can be applied to measure this alignment. We discuss more details in Section 5.

## 4 LONG-FORM VERBALIZED CONFIDENCE

### 4.1 CONFIDENCE ESTIMATION VIA RL

We formulate the task of verbalizing confidence as a sequential decision-making problem on top of language generation. An LLM operates as the policy $\pi_\theta$, parameterized by $\theta$. The objective of the policy is to assign confidence scores to its generated factual statements, such that these scores align with independently verified factuality assessments. Notably, hallucination is not penalized as generation errors; instead, the model is expected to assign low confidence scores to hallucinated statements, thereby facilitating hallucination detection.

We estimate confidence at the *sentence level*, rather than at the passage or atomic-claim level. Sentence-level estimation balances *interpretability, computational efficiency,* and *alignment with natural language structure.* Compared to passage-level estimation, it allows for finer-grained assessment. Compared to atomic-claim-level methods, it avoids extra decomposition steps and produces outputs that are more easily interpreted by humans. Moreover, using numerical confidence scores supports flexible post-processing without affecting text fluency or factual content, unlike methods that embed confidence markers (*e.g.*, "I believe", "I am uncertain") directly into the output. For evaluation, we introduce two task settings for difference use cases: free-form tagging and iterative tagging.

**Free-form Tagging.** We study a setting in which the policy model $\pi_\theta$ produces factual statements along with their associated confidence estimates in a single generation pass. As shown in the left part of Figure 1, in this formulation, the action space includes all possible factual statements $s$ and corresponding confidence values $c$, spanning the model's full vocabulary. The model outputs a sequence of sentence–confidence pairs, $y = \{(s_1, c_1), (s_2, c_2), \dots, (s_n, c_n)\}$, by maximizing the following objective, where $t$ is the $t^{\text{th}}$ output token in sequence $y$:

$$y_t = \operatorname*{argmax}_{y_t} \pi_\theta(y_t | y_{<t}, q) \tag{4}$$

This free-form setting gives the model full generative freedom to balance content generation with calibrated confidence expression. For example, we can use the output confidence to further constrain model to decode only high confidence statements in on-stake domains such as medicine, law, etc.

**Iterative Tagging.** We also evaluate models in a controlled setting where the content is fixed and only the confidence scores are predicted. This setting is motivated by use cases where the generation cannot be altered and provides a consistent basis for model comparison. As shown on the right in Figure 1, given a query $q$ and a base language model $\pi_{\text{base}}$, we first generate a static output $y_{\text{base}} = \{s_1, s_2, \dots, s_n\}$. The policy model $\pi_\theta$ is then asked to assign confidence scores $c_i \in \{0, 1, \dots, 10\}$ for each sentence, conditioned on the query and previously tagged pairs:

$$c_i = \operatorname*{argmax}_{c} \pi_\theta(\{q, (s_1, c_1), (s_2, c_2), \dots, (s_{i-1}, c_{i-1}), s_i\}, c) \tag{5}$$

By decoupling content generation from confidence estimation, this setting ensures fair comparison across models and only requires models to generate confidence scores. In contrast to free-form tagging, it avoids the confounding effects of content variation on confidence evaluation.

**Why RL.** In our study, we prefer RL over SFT for confidence calibration in long-form generation for the following reasons. Standard LLM SFT optimizes likelihood on dense signals from positive references and offers limited leverage from negative samples. Though it learns to assign lower probability to undesirable outputs, but not to adjust the *degree* of confidence or reason about the costs of errors. By contrast, RL is expressly designed for sparse, delayed feedback and can exploit both positive and negative outcomes by directly rewarding alignment between factuality and the emitted confidence score (Kumar et al., 2024; Havrilla et al., 2024). In addition, effective calibration requires *joint* optimization of content and confidence: SFT learns a post-hoc mapping from fixed text to a score, whereas RL treats the sentence *and* its score as one action, enabling credit assignment across both and allowing the model to revise content for better calibration. This also lets us encode ordinal structure and asymmetric penalties (e.g., being confidently wrong is worse than being uncertain) via the reward, without hand-balancing differentiable losses. We propose both on-policy and off-policy training strategies to accommodate different application scenarios.

## 4.2 On-Policy Design

Given a data point $d = (q, E) \sim \mathcal{D}$, containing a query $q$ and the evidence $E$ for verification, the output sequence $y = \{(s_i, c_i)\}_{i=1}^n$ can be sampled from $y = \pi_\theta(q)$. Given an oracle model $\mathcal{O}$, we can obtain the ground truth factuality $\mathbf{f} = \texttt{FactCheck}(\mathcal{O}, q, E, \mathbf{s})$. In our setting, the core design challenge for on-policy RL lies in constructing a reward signal that encourages aligning the model's predicted confidence scores $\mathbf{c}$ with the factual correctness of each statement $\mathbf{f}$.

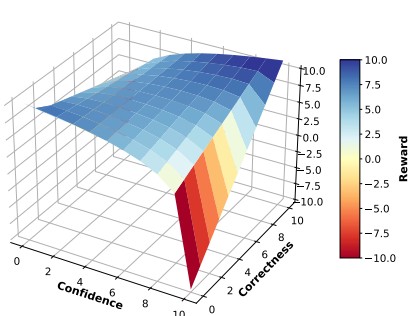

Figure 2: GRPO Reward Function

Intuitively, we want to reward the model when the confidence $c_i$, and correctness $f_i$ for each statement $s_i$ are close (*e.g.*, high correctness - high confidence and vice versa), and penalize the model when they are far part (*e.g.*, low correctness, high confidence, vice versa). Similar to Stangel et al. (2025), we use a log-base reward as it imposes stronger penalties to miscalibration comparing to simple linear and quadratic losses, as visualized in Figure 2. The log-base reward is more appropriate for risk-sensitive applications where confidence must reflect true correctness likelihood. We design this confidence reward $r^{\text{conf}}$ for an output $y$ using binary cross-entropy loss as below, where $\lambda$ is the scaling factor, $R_{max}$ is the normalizing factor and $\odot$ is the Hadamard product.

$$r^{\text{conf}} = \lambda \cdot \frac{1}{n} \mathbf{1}^\top \left( \mathbf{1} + \frac{\mathbf{f} \odot \log(\mathbf{c}) - (\mathbf{1} - \mathbf{f}) \odot \log(\mathbf{1} - \mathbf{c})}{R_{max}} \right) \tag{6}$$

Both confidence $c_i$ and factuality $f_i$ are normalized from integers in $\{0, 1, \ldots, 10\}$ to real numbers in $[0, 1]$ for numerical stability in our implementation. In practice, we combine the confidence reward with other subordinate objectives (*e.g.*, informativeness, format reward) to ensure model is accurately expressing confidence while retaining the quality of generation, with more details in Appendix B.

We instantiate this on-policy setup using the GRPO algorithm (Shao et al., 2024), an on-policy method adapted from PPO (Schulman et al., 2017). Given a dataset $\mathcal{D} = \{d_1, d_2, \ldots, d_N\}$, where each data point $d = (q, E)$, we sample a group of output trajectories $\mathbf{y} = \{y_1, y_2, \ldots, y_G\}$ from the current policy $\pi_{\theta_{\text{old}}}$ and obtain the group reward $\mathbf{r} = \{r_1, \ldots r_G\}$. Then we calculate the averaged advantage $\hat{A}_j(\pi_\theta, \pi_{\text{old}}, y_j, E)$ by computing the reward using fact-checking for policy update. The GRPO loss is defined as below, with $\beta$ as the KL-regularization factor with more details in Appendix B.

$$\mathcal{L}_{\text{GRPO}}(\theta) = - \mathbb{E}_{(q,E) \sim \mathcal{D}, \{y_j\}_{j=1}^G \sim \pi_{\theta_{\text{old}}}(\mathbf{y}|q)} \left[ \frac{1}{G} \sum_{j=1}^G \left( \hat{A}_j(\pi_\theta, \pi_{\text{old}}, y_j, E) - \beta \, \mathbb{D}_{\text{KL}}[\pi_\theta \| \pi_{\text{ref}}] \right) \right] \tag{7}$$

---

**Algorithm 1** Generating Preference Pair Dataset via Fact-Checked Confidence Scores

---

**Require:** Dataset $\mathcal{D} = \{(q_i, E_i)\}_{i=1}^N$, model $\pi_{\text{base}}$, generations per query $n$, orcale model $\mathcal{O}$
**Ensure:** Preference dataset $\mathcal{D}_{\text{pref}} = \{(q_i, y_{w,i}, y_{l,i})\}_{i=1}^N$
1: Initialize $\mathcal{D}_{\text{pref}} \leftarrow \emptyset$
2: **for** each $(q, E) \in \mathcal{D}$ **do**
3:      Generate outputs $y_{\text{base}} = \{s_1, \ldots, s_n\} \leftarrow \pi_{\text{base}}(q)$
4:      Compute *winning* scores $\mathbf{f} = (f_1, \ldots, f_n) \leftarrow \texttt{FactCheck}(\mathcal{O}, q, E, y_{\text{base}})$
5:      Initialize *losing* score vector $\mathbf{c}' = (c_1', \ldots, c_n')$
6:      **for** $j$ from 1 to $n$ **do**                                       ▷ Generate scores for the *losing* example $y_l$
7:          Sample $c_j' \sim \mathcal{U}(\{0, 1, \ldots 10\} \setminus \{f_j\})$          ▷ Random integer in $(\{0, 1, \ldots, 10\} \setminus \{f_j\})$
8:      **end for**
9:      Construct *winning* response set $y_w = \{(s_j, f_j)\}_{j=1}^n$
10:     Construct *losing* response set $y_l = \{(s_j, c_j')\}_{j=1}^n$          ▷ Uses same $s_j$ but different scores $c_j'$
11:     Add preference tuple $(q, y_w, y_l)$ to $\mathcal{D}_{\text{pref}}$
12: **end for**
13: **return** $\mathcal{D}_{\text{pref}}$

---

### 4.3 Off-Policy Design

For off-policy RL, we focus on preference learning Christiano et al. (2017). To construct the preference-pair data $(q, y_w, y_l) \sim \mathcal{D}_{\text{pref}}$, for each query $q$, we need to construct a winning output $y_w$ and a losing output $y_l$. We first probe the model's $\pi_{base}$ initial knowledge, using query $q$ from $(q, E) \in \mathcal{D}$ to elicit the initial response from the model. The response $y_{\text{base}} = \pi_{\text{base}}(q)$ only contains factual statements $y_{\text{base}} = \{s_1, s_2, \ldots, s_n\}$ as we have not taught the model to generate formatted confidence yet. Similarly, we generate the factual correctness score $\mathbf{f} = \texttt{FactCheck}(\mathcal{O}, q, E, y_{base})$ and augment the preference-pair dataset $\mathcal{D}_{\text{pref}}$ for off-policy training, as detailed in Algorithm 1. We use implement DPO Rafailov et al. (2023a) for preference based training . For DPO algorithm, we first finetune the original model for format following on $y_w$ with SFT only, to acquire $\pi_{\text{SFT}}$. We then perform training with the standard DPO objective as below, with $\beta$ to regularize the model's behaviour with respect to the reference model $\pi_{\text{SFT}}$.

$$\mathcal{L}_{\text{DPO}}(\theta) = -\mathbb{E}_{(q, y_w, y_l) \sim \mathcal{D}_{\text{pref}}} \left[ \log \sigma \left( \beta \log \frac{\pi_\theta(y_w \mid q)}{\pi_{\text{SFT}}(y_w \mid q)} - \beta \log \frac{\pi_\theta(y_l \mid q)}{\pi_{\text{SFT}}(y_l \mid q)} \right) \right] \tag{8}$$

## 5 Experiments

### 5.1 Experiments Setup

**Datasets.** We use three datasets for evaluation. Among them, we split WildHallucination (WildHallu) for training and testing, while the other two datasets are used for **testing only**: **(1) WildHallu:** It contains 7919 entities mined from user-chatbot conversations collected in the wild. We divide the original dataset (Zhao et al., 2024) into training, development, and test sets with a 8:1:1 ratio. **(2) Bios**: It consists of 183 human-annotated entities related to people on Wikipedia from FActScore (Min et al., 2023), covering a wide range of popularity levels. It has been widely used for evaluating both long-form factuality and uncertainty (Zhang et al., 2024a;b; Jiang et al., 2024). **(3) PopQA** (Mallen et al., 2023): Following Jiang et al. (2024), we use the long-form version of PopQA, which comprises entities across diverse topics such as people, cities, movies, and companies.

**Fact-checking.** Both Bios and PopQA provide corresponding Wikipedia pages as evidence. For WildHallu, the dataset authors provide the top-10 Google Search results for each entity. During fact-checking, we input the content to be verified alongside the collected evidence, following the pipelines described in (Zhang et al., 2024a; Zhao et al., 2024; Min et al., 2023). Specifically, we use GPT-4o to obtain more accurate judgments. We conduct additional human annotation to double check this pipeline in Appendix C. The detailed prompting strategy is provided in the Appendix M.

**Baselines.** We select baselines according to two key criteria. First, a method must produce a structured, numerical confidence score for each output. This criterion excludes methods that do not generate per-instance scores (Jiang et al., 2024; Kuhn et al., 2023), as well as approaches like LoGU

Table 1: **Free-form tagging** results using Llama3-8B-Instruct. The top three results outperforming LUQ are highlighted in cyan, with deeper shades indicating better performance. All values are presented as percentages.

| Method | WildHallu | | | Bios | | | PopQA | | |
|---|---|---|---|---|---|---|---|---|---|
| | BS↓ | ECE-M↓ | SC↑ | BS↓ | ECE-M↓ | SC↑ | BS↓ | ECE-M↓ | SC↑ |
| Literature SOTA | | | | | | | | | |
| LUQ | 14.5 | 21.5 | 56.8 | 20.0 | 29.5 | 63.8 | 16.7 | 23.2 | 62.5 |
| Our Methods | | | | | | | | | |
| LoVeC-SFT | 8.9 | 15.1 | 58.8 | 16.6 | 26.1 | 58.9 | 19.4 | 27.8 | 52.6 |
| LoVeC-GRPO | 6.0 | 8.2 | 63.1 | 10.1 | 11.1 | 68.7 | 10.1 | 5.1 | 63.0 |
| LoVeC-DPO | 6.3 | 5.4 | 62.1 | 9.2 | 6.1 | 67.4 | 10.3 | 4.0 | 62.6 |

Table 2: **Iterative tagging** results using Llama3-8B-Instruct. The top three results outperforming LUQ are highlighted in cyan, with deeper shades indicating better performance. All values are presented as percentages.

| Method | WildHallu | | | Bios | | | PopQA | | |
|---|---|---|---|---|---|---|---|---|---|
| | BS↓ | ECE-M↓ | SC↑ | BS↓ | ECE-M↓ | SC↑ | BS↓ | ECE-M↓ | SC↑ |
| Literature SOTA | | | | | | | | | |
| LUQ | 14.5 | 21.5 | 56.8 | 20.0 | 29.5 | 63.8 | 16.7 | 23.2 | 62.5 |
| Baseline Methods | | | | | | | | | |
| Vanilla | 10.8 | 6.0 | 9.1 | 20.9 | 24.1 | 1.2 | 21.7 | 23.7 | 4.9 |
| p(true) | 23.8 | 23.6 | 15.8 | 19.7 | 28.6 | 17.3 | 19.9 | 24.3 | 23.1 |
| Verb-Conf | 20.3 | 22.1 | 13.4 | 21.2 | 25.3 | 10.8 | 18.8 | 22.1 | 18.3 |
| Self-Cons | 16.5 | 24.3 | 47.8 | 20.3 | 26.5 | 58.8 | 17.3 | 21.6 | 56.8 |
| Our Methods | | | | | | | | | |
| LoVeC-SFT | 9.1 | 15.2 | 51.1 | 16.6 | 25.8 | 56.0 | 18.0 | 25.9 | 52.7 |
| LoVeC-GRPO | 5.7 | 2.5 | 57.0 | 8.5 | 4.2 | 64.7 | 11.3 | 6.2 | 62.8 |
| LoVeC-DPO | 6.0 | 5.0 | 60.4 | 9.0 | 7.3 | 65.6 | 9.6 | 1.7 | 63.1 |

(Yang et al., 2024) and Linguistic Calibration (Band et al., 2024), which embed natural language uncertainty phrases that are not suitable for automated quantitative comparison. Second, the method must operate at the sentence level, without requiring fine-grained atomic claim decomposition using GPTs (Fadeeva et al., 2024; Liu et al., 2024). Prompt formulations for all baselines are provided in Appendix M.

- **Vanilla**: This refers to directly prompting the original model (*e.g.*, Llama-3-8B-Instruct).
- **p(true)** (Kadavath et al., 2022): We present a sentence to an LLM and ask whether it is factually true or false. The likelihood associated with the "true" label is used as the confidence score. Following (Zhang et al., 2024b), we provide additional context to the LLM to address co-reference issues.
- **Verbalized Confidence (Verb-Conf)** (Xiong et al., 2024; Tian et al., 2023): We prompt the LLM to assign a numerical confidence score (ranging from 0 to 10) to a given sentence, reflecting the model's belief in its factuality. Similar to p(true), we additionally provide the full paragraph as context to the model.
- **Self-Consistency (Self-Cons)** (Manakul et al., 2023): We generate 10 additional outputs using temperature $T = 1$ and compute the agreement between the original output and the sampled outputs. The level of agreement is used as the confidence score.
- **LUQ** (Zhang et al., 2024a): A state-of-the-art (SOTA) uncertainty estimation method specifically designed for long-form QAs. LUQ demonstrates better performance over a range of baselines in short-form uncertainty estimation (Lin et al., 2023; Kuhn et al., 2023) and is also applied to confidence estimation.

**Training Settings.** For the backbone language models, we use Llama-3-8B-Instruct (Meta, 2024) and Gemma-2-9B-It (Team et al., 2024). We first perform one epoch of SFT on $y_w$ from the **Wildhallu**

preference dataset for format adherence. For a fair comparison, we subsequently fine-tune each model for one additional epoch using SFT, GRPO, DPO, respectively. For GRPO, we use a copy of the model itself as reward model for online reward assignment. More training details are in Appendix B.

**Evaluation Metrics**  Since both factuality and confidence lie in $[0, 1]$, we use metrics suited to continuous labels: **(1) Brier Score (BS)** for mean squared error between predicted confidence and correctness, **(2) ECE-M** (Huang et al., 2024a) for calibration under soft labels, and **(3) Spearman Correlation (SC)** (Zhang et al., 2024a) to assess ordinal consistency. All results use greedy decoding.

## 5.2 EXPERIMENTAL RESULTS

**LoVeC demonstrates substantial improvement on calibration in both freeform and iterative tagging.** As shown in Tables 1 and 2, `LoVeC-DPO` and `LoVeC-GRPO` consistently outperform all baselines, including the prior SOTA LUQ, across all evaluation metrics. This trend holds for both Llama and Gemma. While SFT alone achieves results comparable to some baselines, applying RL further improves performance, highlighting the necessity of optimizing confidence via RL. As depicted in Table 7 and 8 (Appendix D), by averaging sentence-level confidence and factuality over generated passage, the results exhibit consistent trends in passage-level. Additional studies in Appendix F confirms our models' confidence is directly associate to the current fact during generation, and not affected by previously assigned confidence scores. A case study can be found in Appendix L.

**LoVeC is highly efficient on test-time.** Our method offers the btter test-time efficiency. Confidence scores are generated inline with the answer, requiring no additional sampling or decomposition of responses into atomic claims via external API calls. In contrast, existing state-of-the-art sampling-based methods—such as LUQ for long-form generation—incur significant overhead due to repeated sampling and similarity computations. As depicted in Figure 3 our method completes the inference on **Wildhallu** test set (792 instances) 20 times faster than existing SOTA LUQ on free-form tagging. A detailed discussion of the underlying reasons for this efficiency is provided in Appendix K.

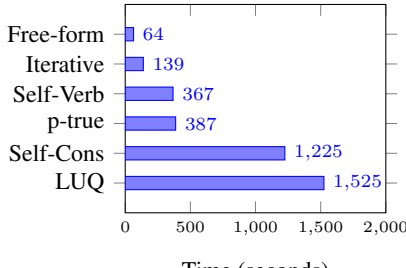

Figure 3: Running-time Comparison

**LoVeC generalizes well across domains and short-form QA.** Tables 1 and 2 show that `LoVeC` generalizes effectively to diverse datasets such as Bios and PopQA. To assess cross-format transfer, we test the model's ability to adapt to short-form confidence estimation using the TriviaQA dataset (Joshi et al., 2017), a benchmark for short-form QA. As shown in Table 20 and Appendix J), our RL-trained models achieve competitive ECE and AUROC scores compared to the baselines. Notably, `LoVeC` approaches the performance of the state-of-the-art RL-based method, RewardingDoubt (Stangel et al., 2025), despite being trained on significantly less and fully out-of-domain data. More details are in Appendix J. Overall, the results highlight the robustness and transferability of `LoVeC` across both domains and task formats.

**LoVeC preserves response length and overall factuality.** In the freeform tagging, our RL-trained models may produce different content compared to the original model. We further compares the generation lengths and factuality. `LoVeC` maintains both response length and factual accuracy, confirming that our calibration improvements do not compromise informativeness and showing no signs of reward hacking. Full details are in Appendix H.

## 6 ANALYSIS

**RL ensures numerical consistency.** Examining the top-ranked tokens shows that RL-trained models, especially GRPO, assign probabilities that respect the ordinal structure of the confidence scale. As seen in Table 3, for RL methods, higher scores (*e.g.*, 10, 9, 8) reliably outrank lower ones in factually correct generation. Even under factually incorrect case (*i.e.*, model hallucinates about an unknown fact) RL methods maintains an ordered distribution centered on its prediction. Tokens representing higher confidence appear in monotonic order with decreasing probability (*e.g.*, 3, 4 comes after 2, and

Table 3: Case study on predicting the next confidence score token. We use one factually **Correct** sentence and one **Incorrect** sentence. The table lists the top-15 tokens; unrenderable characters are shown as [?], and spaces are displayed as ␣. GRPO exhibits a clear ordinal pattern, DPO shows partial ordering, and SFT shows little to none. See Appendix I for the prompt.

| Model | Top 15 Tokens | | | | | | | | | | | | | | |
|---|---|---|---|---|---|---|---|---|---|---|---|---|---|---|---|
| **Correct**: King's College, Cambridge is a constituent college ... and most prestigious universities. <confidence> | | | | | | | | | | | | | | | |
| GRPO | 10 | 9 | 8 | 7 | 6 | 5 | 4 | 3 | 2 | 1 | 0 | ␣ | 11 | 90 | 99 |
| DPO | 10 | 9 | 8 | 7 | 11 | 6 | 5 | [?] | 09 | ␣tenth | [?] | 12 | 4 | [?] | ␣ten |
| SFT | 10 | 0 | 1 | 4 | 8 | 2 | 3 | 7 | 11 | 9 | 5 | 6 | 12 | ␣X | X |
| **Incorrect**: MiniGPT4 is a lightweight and efficient variant of ... in resource-constrained environments. <confidence> | | | | | | | | | | | | | | | |
| GRPO | 2 | 3 | 4 | 5 | 1 | 6 | 0 | 7 | 8 | 9 | 10 | ␣ | 30 | 20 | 60 |
| DPO | 2 | 3 | 4 | 5 | 6 | 1 | 7 | 0 | 8 | 9 | 10 | [?] | ␣five | four | ␣four |
| SFT | 0 | 10 | 1 | 4 | 2 | 3 | 8 | 7 | 5 | 6 | 9 | 11 | 12 | 13 | 14 |

0 comes after 1). We believe such desired behavior stems from the RL reward. GRPO shows the best ordering since its reward explicitly aligns confidence with factuality. DPO exhibits partial ordering but is often disrupted by irrelevant tokens, reflecting weaker ordinal constraints. SFT performs worst: despite outputting plausible top scores (e.g., 10), subsequent tokens lack meaningful order, with anomalies like 0 ranked highly. This lack of structural supervision undermines calibration. More details are in Appendix I.

**Ablating the oracle model achieves on-par results.** For DPO, we initially employ GPT-4o as an oracle model to generate preference pairs based on factuality comparisons. To assess the necessity of this external supervision, we perform an ablation study by replacing GPT-4o with a self-labeling setup. For instance, Llama-3-8B-Instruct generated outputs are fact-checked using a frozen copy of itself. Our GRPO pipeline is oracle-free by design, as generating GPT-4o labels online during training is prohibitively expensive. As shown in Appendix D.3, Table 9, DPO trained with self-generated labels performs slightly worse than those using GPT-4o, but still outperforms the strongest baseline, LUQ. The success of self-labeling highlights the potential for scalability in settings where external oracle models are unavailable.

**GRPO reward design improves calibration, while SFT regression offers no gains..** For GRPO, we further examine the impact of alternative reward formulations. In addition to the log-based reward in Equation 6, we experiment with linear and quadratic variants based on the absolute and squared difference between predicted confidence and correctness as the target of alignment. As shown in Table 10, all reward functions promote such alignment, but the log-based reward proves more effective: as a proper scoring rule, it sharply penalizes overconfident errors and provides stronger calibration. We also explore whether replacing cross-entropy with a regression loss on confidence scores during SFT improves calibration. However, as reported in Appendix G, this modification yields no benefit, further confirming the inherent limitations and inefficiency of SFT for this task.

**Suggestions to Practitioners.** Both RL methods deliver strong and reliable performance, but with distinct trade-offs. GRPO, though more computationally intensive due to its explicit reward model, offers key advantages: it directly models ordinal relationships between confidence scores and provides improved numerical consistency. In contrast, DPO avoids deploying a separate reward model but relies on carefully curated offline preference pairs, which can be costly to construct and may restrict flexibility. Thus, GRPO is preferable when ample computational resources are available, while DPO serves as a lighter-weight alternative under tighter resource constraints.

# 7 CONCLUSION

We introduce `LoVeC`, a reinforcement learning method to improve confidence estimation in long-form factual text generation. Our approach achieves SOTA performance in both confidence calibration and runtime efficiency. Our results also demonstrate that RL enables more consistent and interpretable confidence predictions. Further analysis shows strong generalization and scalability of our model to out-of-domain datasets and short-form confidence estimations. The results highlight the potential of our framework for deployment in risk-sensitive and high-stakes domains, or general LLM use cases, where hallucination detection is crucial for trust and usability.

## REPRODUCIBILITY STATEMENT

Datasets we used, WildHallucinations, Bios, PopQA, and TriviaQA, are all publicly available. Prompts used are fully described in Appendix M. We use publicly released backbones (Llama-3-8B-Instruct, Gemma-2-9B-It). All fact checking pipelines and human annotation protocols are described in Appendix C. Training scripts, configuration files, will be released as anonymous supplementary material. Full reward formulations and GRPO loss equations are provided in Appendix B, together with hyperparameters, optimization settings, and LoRA configurations. Experiments were run on Google Cloud A100 80GB GPUs ( 1500 GPU hours). Software stack and licenses are listed in Appendix B.

## ETHICS STATEMENT

Our research adheres to the ICLR Code of Ethics. We do not foresee any risks or potential harm from this study. All datasets and code used are under appropriate licenses. Human annotation was conducted following standard practices, with annotators providing consent to share the data for research purposes. We use LLMs only for polishing the paper writing.

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

Table 4: On the advantage of verbalized confidence. We compare four paradigms along three axes: efficiency, suitability for long-form generation, and flexibility/machine-interpretability. Our numerical verbalization consolidates the strengths of prior paradigms: it is efficient, works well for long-form outputs, and is easy to parse/threshold.

| Paradigm | Efficient? | Suitable for Long-form? | Flexibility & Machine Interpretability |
|---|---|---|---|
| Sampling/Consistency-based | ✗ | ✓ | ✓ |
| Logit/Probability-based | ✓ | ✗ | ✓ |
| Verbalized Confidence (Linguistic) | ✓ | ✓ | ✗ |
| Verbalized Confidence (Numerical) — Ours | ✓ | ✓ | ✓ |

## A  LIMITATION AND FUTURE WORK

Our reinforcement learning tuning approach requires access to white-box models, which limits its applicability to black-box settings. Another limitation is our exclusive focus on factuality; this choice is guided by the availability of widely adopted long-form factuality evaluation pipelines in existing research. Future work could explore several directions. First, confidence estimation can be extended to more general long-form generation tasks such as code generation, creative writing, and machine translation. Second, applying our method to high-stakes domains—such as law, healthcare, and finance—represents an important and impactful avenue for future research.

**Broader Impact**   Our work presents potential for enhancing the trustworthiness of large language models in real-world deployments, especially in high-stakes domains such as healthcare, law, and education, by improving sentence-level confidence estimation and reducing hallucinations. The interpretability and efficiency of our method may enable safer AI systems by allowing users to make informed decisions based on model-generated content. However, we recognize that verbalized confidence could be misused—for example, to give unwarranted credibility to inaccurate outputs or manipulate perceived authority. As such, careful deployment and transparency about confidence generation mechanisms are essential to prevent misuse and ensure ethical adoption.

## B  EXPERIMENT DETAILS

### B.1  TRAINING SETUP

In our experiment we use SFT , GRPO, DPO, and ORPO. We choose them also as an ablation of reward and reference model, with the details in Tabel 5 below. We design a confidence quantification

| Method | Reward Model | Reference Model |
|---|---|---|
| GRPO | Yes | Yes |
| DPO | No | Yes |
| ORPO | No | No |

Table 5: Comparison of methods by use of Reward Model and Reference Model

prompt for instruction-following, which is prepended before each query. However, we observe that the models often fail to generate responses that follow the expected confidence format. Thus, for both of `Llama-3-8B-Instruct` and `Gemma-2-9b-it`, we perform 1 epoch of SFT on $(q, y_w) \sim \mathcal{D}_{\text{pref}}$ for format following on the completion $y_w$ only before RL. For GRPO, we use the frozen copy of original model as the reward model for fact-checking. Both DPO and ORPO are using the exact same $\mathcal{D}_{\text{pref}}$.

We use LoRA Hu et al. (2022) on `q_proj`, `k_proj`, `v_proj`, `o_proj` consistently across models and methods to fine-tune $< 1\%$ of the model's parameters. Below are the detailed hyperparameter choices.

## B.2 INFRASTRUCTURE

We've used TRL von Werra et al. (2020) libraries for training and vLLM Kwon et al. (2023) libraries for inference. We've conducted our experiments on Google Cloud Platform using `a2-ultragpu` machines with `A100 80GB` GPUs. We have consumed around 1500 GPU hours for this project. We list the assets we used and their license in Table 6.

| Asset | Category | License |
|---|---|---|
| TRL v0.15.2 | Code | Apache License 2.0 |
| vLLM v0.7.3 | Code | Apache License 2.0 |
| WildHallucinations | Dataset | MIT License |
| Bios | Dataset | MIT License |
| PopQA | Dataset | MIT License |
| TriviaQA | Dataset | Apache License 2.0 |

Table 6: List of external assets used and their licenses.

## B.3 GRPO LOSS DESIGN

Here we provide the full equation of our GRPO loss. For each data point $d = (q, E)$ a dataset $\mathcal{D} = \{d_1, d_2, \ldots, d_N\}$, we sample a group of output trajectories $\mathbf{y} = \{y_1, y_2, \ldots, y_G\}$ from the current policy $\pi_{\theta_{\text{old}}}$ and obtain the group reward $\mathbf{r} = \{r_1, \ldots r_G\}$. Then we optimize a new policy $\pi_\theta$ based on the per-output advantage estimates $\forall y_j \in \mathbf{y}, \hat{A}_j = \frac{r_j - \text{mean}(\mathbf{r})}{\text{std}(\mathbf{r})}$. $\epsilon$ is the clipping factor which helps to stabilize training by preventing excessively large policy updates. $\pi_{ref}$ is the model's frozen copy for KL-regularization. $t$ denotes the $t^{th}$ token of trajectory $y$.

$$
\mathcal{L}_{\text{GRPO}}(\theta) = -\mathbb{E}_{q \sim \mathcal{D}, \{y_j\}_{j=1}^G \sim \pi_{\theta_{\text{old}}}(\mathbf{y}|q)} \frac{1}{G} \sum_{j=1}^G \frac{1}{|y_j|} \sum_{t=1}^{|y_j|} \left\{ \min \left[ \frac{\pi_\theta(y_{j,t} \mid q, y_{j,<t})}{\pi_{\theta_{\text{old}}}(y_{j,t} \mid q, y_{j,<t})} \hat{A}_{j,t}, \right. \right.
$$
$$
\left. \left. \text{clip} \left( \frac{\pi_\theta(y_{j,t} \mid q, y_{j,<t})}{\pi_{\theta_{\text{old}}}(y_{j,t} \mid q, y_{j,<t})}, 1-\epsilon, \, 1+\epsilon \right) \hat{A}_{j,t} \right] - \beta \, \mathbb{D}_{\text{KL}}[\pi_\theta \| \pi_{\text{ref}}] \right\}
\tag{9}
$$

In our implementation, we applied reward stretching to make sure it is sensitive enough to model's responses. In order to retain the quality of model's generation, we additionally added subordinate rewards, $r^{correct}$ represents the total factuality score, judged by reward model. The python implementation of our reward function is below.

```python
def improved_log_reward(confidence: int, correctness: int,
scale=10.0, gamma=1.5, penalty_strength=5.0):
    if confidence is None or not (0 <= confidence <= 10):
        return -3 * scale  # malformed input penalty

    # Core log-likelihood reward
    p = np.clip(confidence / 10, 1e-6, 1 - 1e-6)
    y = correctness / 10
    nll = -(y * math.log(p) + (1 - y) * math.log(1 - p))

    best_nll = 0
    worst_nll = -(math.log(1e-6) + math.log(1 - 1e-6)) / 2

    reward = scale * (1 - (nll - best_nll) / (worst_nll - best_nll))

    # Stretch reward to amplify good/bad
    reward = np.sign(reward) * (abs(reward) ** gamma)

    # Correctness bonus (small)
    reward += 0.15 * correctness

    return float(reward)
```

## C    HUMAN ANNOTATION: RELIABILITY OF GPT-4O ANNOTATIONS

Although the paradigm of using GPT+Evidence to fact-check has been widely used in previous work (Zhang et al., 2024a; Zhao et al., 2024; Min et al., 2023; Wei et al., 2024), we conduct additional human annotation to evaluate the reliability of using GPT-4o as a fact-checker with retrieved evidence. Two annotators with strong English proficiency and a master's degree in computer science were recruited. They were instructed to fact-check the sentences using the same prompt provided to GPT-4o. A random sample of 50 instances was drawn from the WildHallu dataset, consisting of 312 sentences in total. We use Spearman correlation as the metric for reliability assessment. The inter-annotator agreement is 0.91 between Annotator 1 and Annotator 2. For the comparison between GPT-4o and the human average, we observe a Spearman correlation of 0.88, indicating a very strong alignment between the model and human judgments.

## D    ADDITIONAL RESULTS ON LLAMA-3-8B-INSTRUCT

### D.1    RELIABILITY DIAGRAMS

Figure 4 displays reliability diagrams for the SOTA method LUQ, the vanilla model, `LoVeC-SFT`, and `LoVeC-DPO`. A reliability curve closer to the perfect calibration line signifies better calibration. We observe that both the vanilla model and `LoVeC-SFT` exhibit severe overconfidence. In contrast, our `LoVeC-DPO` method breaks this overconfidence pattern, leading to improved calibration results.

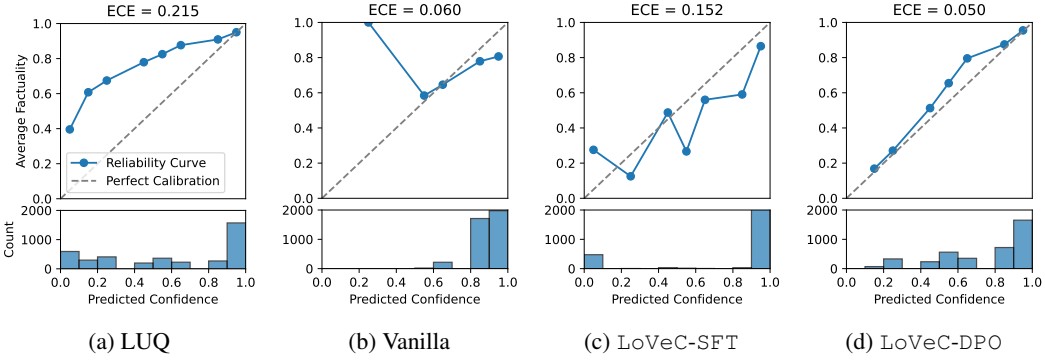

(a) LUQ          (b) Vanilla          (c) LoVeC-SFT          (d) LoVeC-DPO

Figure 4: Reliability diagrams for iterative tagging using Llama3-8B-Instruct in sentence-level.

### D.2    PASSAGE-LEVEL RESULTS

We provide passage-level results of `Llama-3-8B-Instruct`. We simply estimate the passage-level performance by calculating the average of sentence-level confidence and factuality. As shown in the tables below, our method, `LoVeC`, provides better performance than literature SOTA.

### D.3    ABLATING THE ORACLE MODEL

To assess the necessity of using a high-capacity oracle model, we conduct an ablation study by replacing GPT-4o with Llama-3-8B-Instruct for generating preference datasets. Specifically, instead of relying on GPT-4o for fact-checking and labeling preference pairs, we use the training model itself to self-label its outputs prior to DPO training.

As shown in Table 9, while models trained on self-labeled data perform slightly worse than those using GPT-4o supervision, they still surpass strong baselines. Notably, `LoVeC-DPO` trained with self-labeling continues to outperform the previous state-of-the-art method, LUQ. This result highlights the practicality and effectiveness of oracle-free training, making the approach more accessible and cost-efficient without significantly compromising performance.

Table 7: Passage-level iterative tagging results using Llama3-8B-Instruct. The top three results outperforming LUQ are highlighted in cyan, with deeper shades indicating better performance. All values are presented as percentages.

| Method | WildHallu | | | Bios | | | PopQA | | |
|---|---|---|---|---|---|---|---|---|---|
| | BS↓ | ECE-M↓ | SC↑ | BS↓ | ECE-M↓ | SC↑ | BS↓ | ECE-M↓ | SC↑ |
| Literatrue SOTA | | | | | | | | | |
| LUQ | 8.0 | 19.1 | 70.5 | 12.4 | 28.1 | 75.3 | 9.8 | 21.9 | 73.1 |
| Baseline Methods | | | | | | | | | |
| Vanilla | 8.4 | 7.1 | 30.0 | 17.4 | 26.0 | 18.5 | 18.9 | 26.0 | 18.3 |
| p(true) | 15.4 | 19.9 | 27.6 | 16.2 | 23.8 | 29.4 | 17.3 | 19.6 | 34.1 |
| Verb-Conf | 17.9 | 20.3 | 23.8 | 17.7 | 20.1 | 22.7 | 16.4 | 16.8 | 25.4 |
| Self-Cons | 12.1 | 17.4 | 59.2 | 18.3 | 21.2 | 64.7 | 14.7 | 17.1 | 61.3 |
| Our Methods | | | | | | | | | |
| LoVeC-SFT | 6.8 | 15.5 | 54.0 | 13.1 | 26.6 | 63.9 | 14.4 | 26.3 | 60.5 |
| LoVeC-GRPO | 3.3 | 2.7 | 72.5 | 5.0 | 5.0 | 77.2 | 7.7 | 7.5 | 75.1 |
| LoVeC-DPO | 3.5 | 5.5 | 73.1 | 5.2 | 7.0 | 78.3 | 6.1 | 5.5 | 74.1 |

Table 8: Sentence-level free-form tagging results using Llama3-8B-Instruct. The top three results outperforming LUQ are highlighted in cyan, with deeper shades indicating better performance. All values are presented as percentages.

| Method | WildHallu | | | Bios | | | PopQA | | |
|---|---|---|---|---|---|---|---|---|---|
| | BS↓ | ECE-M↓ | SC↑ | BS↓ | ECE-M↓ | SC↑ | BS↓ | ECE-M↓ | SC↑ |
| Literature SOTA | | | | | | | | | |
| LUQ | 14.5 | 21.5 | 56.8 | 20.0 | 29.5 | 63.8 | 16.7 | 23.2 | 62.5 |
| Our Methods | | | | | | | | | |
| LoVeC-SFT | 6.4 | 15.2 | 60.6 | 12.4 | 25.6 | 66.1 | 15.4 | 27.1 | 59.5 |
| LoVeC-GRPO | 3.8 | 8.0 | 73.0 | 5.8 | 10.8 | 81.5 | 6.4 | 5.3 | 73.6 |
| LoVeC-DPO | 3.6 | 5.6 | 73.1 | 5.3 | 6.9 | 78.1 | 6.5 | 3.9 | 73.2 |

## D.4 THE SELECTION OF REWARD FUNCTION

We compare different reward functions used in our GRPO framework, including logarithmic, linear, and quadratic forms, as shown in Table 10. This demonstrates that the choice of reward function plays a crucial role in guiding the learning process.

## E GEMMA-2-9B-IT RESULTS

Here we provide results for Gemma-2-9B-It. As described in the tables below, our method shows consistent improvements across models, beating the literature SOTA, LUQ, across datasets.

Table 9: Comparison of WildHallu results for LUQ, SFT, DPO, and GRPO across tagging strategies. All values are presented as percentages.

| Method | Iterative Tagging | | | Freeform Tagging | | |
|---|---|---|---|---|---|---|
| | BS | ECE-M | SC | BS | ECE-M | SC |
| LUQ | 14.5 | 21.5 | 56.8 | 14.5 | 21.5 | 56.8 |
| LoVeC-GRPO | 5.7 | 2.5 | 57.0 | 6.0 | 8.2 | 63.1 |
| **Fact-Checking with GPT-4o + Evidence** | | | | | | |
| LoVeC-SFT | 9.1 | 15.2 | 51.1 | 8.9 | 15.1 | 58.8 |
| LoVeC-DPO | 6.0 | 5.0 | 60.4 | 6.3 | 5.4 | 62.1 |
| **Fact-Checking with Llama3-8B + Evidence** | | | | | | |
| LoVeC-SFT | 8.2 | 9.0 | 49.6 | 8.3 | 9.5 | 58.3 |
| LoVeC-DPO | 7.2 | 9.8 | 58.0 | 7.1 | 7.8 | 60.4 |

Table 10: Comparison of WildHallu results for LUQ, GRPO-log, GRPO-linear, GRPO-quadratic. All values are presented as percentages.

| Method | Iterative Tagging | | | Freeform Tagging | | |
|---|---|---|---|---|---|---|
| | BS | ECE-M | SC | BS | ECE-M | SC |
| LUQ | 14.5 | 21.5 | 56.8 | 14.5 | 21.5 | 56.8 |
| GRPO-log | 5.7 | 2.5 | 57.0 | 6.0 | 8.2 | 63.1 |
| GRPO-quadratic | 7.0 | 8.7 | 55.1 | 7.3 | 9.3 | 62.3 |
| GRPO-linear | 8.5 | 10.8 | 54.3 | 8.2 | 10.4 | 59.8 |

Table 11: Sentence-level iterative tagging results using Gemma-2-9B-It. The top three results outperforming LUQ are highlighted in cyan, with deeper shades indicating better performance. All values are presented as percentages.

| Method | WildHallu | | | Bios | | | PopQA | | |
|---|---|---|---|---|---|---|---|---|---|
| | BS↓ | ECE-M↓ | SC↑ | BS↓ | ECE-M↓ | SC↑ | BS↓ | ECE-M↓ | SC↑ |
| **Literature SOTA** | | | | | | | | | |
| luq | 11.9 | 16.3 | 50.0 | 12.2 | 15.5 | 69.2 | 13.6 | 15.1 | 62.6 |
| **Our Methods** | | | | | | | | | |
| Vanilla | 22.5 | 26.3 | 28.9 | 24.5 | 28.8 | 35.5 | 24.0 | 27.7 | 23.7 |
| p(true) | 19.3 | 22.8 | 25.4 | 21.0 | 25.0 | 31.0 | 21.5 | 24.5 | 26.0 |
| Verb-Conf | 18.5 | 19.2 | 35.1 | 18.0 | 19.0 | 39.5 | 19.5 | 20.0 | 36.2 |
| Self-Cons | 13.4 | 17.7 | 43.2 | 13.0 | 17.0 | 53.0 | 13.5 | 16.5 | 48.5 |
| **Our Methods** | | | | | | | | | |
| LoVeC-SFT | 8.0 | 12.2 | 36.1 | 18.8 | 25.1 | 54.9 | 25.8 | 32.6 | 37.1 |
| LoVeC-GRPO | 7.3 | 5.6 | 52.2 | 10.7 | 11.1 | 72.4 | 13.1 | 18.1 | 64.2 |
| LoVeC-DPO | 4.1 | 1.3 | 51.8 | 7.5 | 7.3 | 75.2 | 11.6 | 13.2 | 65.3 |

Table 12: Passage-level iterative tagging results using Gemma-2-9B-It. The top three results outperforming LUQ are highlighted in cyan, with deeper shades indicating better performance. All values are presented as percentages.

| Method | WildHallu | | | Bios | | | PopQA | | |
|---|---|---|---|---|---|---|---|---|---|
| | BS↓ | ECE-M↓ | SC↑ | BS↓ | ECE-M↓ | SC↑ | BS↓ | ECE-M↓ | SC↑ |
| Literature SOTA | | | | | | | | | |
| LUQ | 6.3 | 14.1 | 61.2 | 6.8 | 14.2 | 81.6 | 9.2 | 11.8 | 73.8 |
| Baseline Methods | | | | | | | | | |
| Vanilla | 19.3 | 23.1 | 35.5 | 21.6 | 29.4 | 43.1 | 21.4 | 28.1 | 33.6 |
| p(true) | 17.8 | 20.3 | 28.7 | 19.6 | 24.2 | 33.8 | 20.4 | 23.7 | 30.2 |
| Verb-Conf | 16.3 | 17.1 | 34.6 | 17.5 | 17.9 | 35.7 | 18.4 | 18.8 | 32.3 |
| Self-Cons | 12.2 | 15.4 | 48.9 | 12.9 | 16.1 | 58.1 | 13.4 | 15.2 | 54.4 |
| Our Methods | | | | | | | | | |
| LoVeC-SFT | 6.5 | 12.1 | 31.9 | 16.2 | 25.6 | 55.6 | 23.6 | 32.8 | 38.3 |
| LoVeC-GRPO | 2.9 | 3.6 | 64.4 | 3.4 | 5.7 | 82.5 | 8.6 | 7.5 | 74.3 |
| LoVeC-DPO | 2.5 | 2.9 | 65.9 | 4.6 | 7.8 | 84.3 | 9.0 | 13.2 | 75.4 |

Table 13: Sentence-level free-form tagging results using Gemma-2-9B-It. The top three results outperforming LUQ are highlighted in cyan, with deeper shades indicating better performance. All values are presented as percentages.

| Method | WildHallu | | | Bios | | | PopQA | | |
|---|---|---|---|---|---|---|---|---|---|
| | BS↓ | ECE-M↓ | SC↑ | BS↓ | ECE-M↓ | SC↑ | BS↓ | ECE-M↓ | SC↑ |
| Literature SOTA | | | | | | | | | |
| LUQ | 11.9 | 16.3 | 50.0 | 12.2 | 15.5 | 69.2 | 13.6 | 15.1 | 62.6 |
| Our Methods | | | | | | | | | |
| LoVeC-SFT | 7.3 | 11.6 | 57.2 | 11.5 | 17.8 | 70.6 | 19.4 | 26.3 | 50.4 |
| LoVeC-GRPO | 4.3 | 4.6 | 56.1 | 8.3 | 3.7 | 72.6 | 8.5 | 8.9 | 63.5 |
| LoVeC-DPO | 4.5 | 2.2 | 55.2 | 6.6 | 4.2 | 70.3 | 9.5 | 8.2 | 66.6 |

Table 14: Passage-level free-form tagging results using Gemma-2-9B-It. The top three results outperforming LUQ are highlighted in cyan, with deeper shades indicating better performance. All values are presented as percentages.

| Method | WildHallu | | | Bios | | | PopQA | | |
|---|---|---|---|---|---|---|---|---|---|
| | BS↓ | ECE-M↓ | SC↑ | BS↓ | ECE-M↓ | SC↑ | BS↓ | ECE-M↓ | SC↑ |
| Literature SOTA | | | | | | | | | |
| LUQ | 6.3 | 14.1 | 61.2 | 6.8 | 14.2 | 81.6 | 9.2 | 11.8 | 73.8 |
| Our Methods | | | | | | | | | |
| LoVeC-SFT | 6.0 | 11.7 | 52.3 | 8.3 | 17.6 | 73.9 | 16.0 | 25.7 | 55.8 |
| LoVeC-GRPO | 3.3 | 4.0 | 66.3 | 3.8 | 4.7 | 83.6 | 6.2 | 4.3 | 78.2 |
| LoVeC-DPO | 2.7 | 3.2 | 65.4 | 3.1 | 4.6 | 83.9 | 6.1 | 7.9 | 77.4 |

## F  SHOULD THE MODEL SEE PREVIOUSLY TAGGED LABELS?

Our iterative tagging protocol conditions each sentence's confidence on all previously tagged (sentence, score) pairs. A natural concern is whether this *sequential conditioning* introduces bias. We therefore ablate visibility of prior scores and compare against the default setting where the model does see them.

**Setup.**  We keep the model (Llama3-8B-Instruct), data split (WildHallu), verifier, and decoding identical to the main iterative tagging experiments and only change the input format:

> **Original (with prior scores):** $\{s_1, c_1, s_2\} \to c_2$,  **No previous scores:** $\{s_1, s_2\} \to c_2$.

Concretely, at step $i$ the Original setting conditions on $(q, (s_1, c_1), \ldots, (s_{i-1}, c_{i-1}), s_i)$ (Eq. (5) in the main paper), whereas the No-Previous-Scores variant conditions on $(q, s_{i-1}, s_i)$ but *omits* $\{c_1, \ldots, c_{i-1}\}$. All other details follow the iterative tagging evaluation in the main text.

| Setting | Method | BS $\downarrow$ | ECE-M $\downarrow$ | SC $\uparrow$ |
|---|---|---|---|---|
| Original | LoVeC-GRPO | 5.7 | 2.5 | 57.0 |
| No Previous Scores | LoVeC-GRPO | 8.1 | 4.4 | 43.0 |
| Original | LoVeC-DPO | 6.0 | 5.0 | 60.4 |
| No Previous Scores | LoVeC-DPO | 7.2 | 6.2 | 52.3 |

Table 15: Effect of hiding previous confidence labels in iterative tagging.

**Findings.**  Hiding previously tagged labels degrades all metrics for both training schemes. Our hypothesis is that the prior score acts as a local calibration anchor that helps the model focus its uncertainty estimate on the *current* sentence rather than implicitly re-evaluating the **entire prefix**. Removing that anchor consistently harms calibration with a stronger effect under GRPO.

These results provide *no* evidence that sequential conditioning introduces a harmful bias. On the contrary, allowing the model to see previously tagged labels yields materially better calibration and discrimination. We therefore recommend *including* prior scores for iterative tagging; the No-Previous-Scores variant remains a viable ablation, but it incurs substantial performance loss.

## G  DO WE NEED REGRESSION LOSS IN SFT?

We evaluate whether using a regression loss to SFT improves confidence estimation under our iterative tagging protocol (We use **Llama-3-8B-Instruct** on **WildHallu** as example).

Table 16: **SFT vs. SFT-regression under iterative tagging.** Lower is better for BS/ECE-M; higher is better for AUROC.

| Model | BS $\downarrow$ | ECE-M $\downarrow$ | SP $\uparrow$ |
|---|---|---|---|
| LoVeC-SFT | 9.1 | 15.2 | 51.1 |
| LoVeC-SFT-regression | 12.9 | 19.8 | 47.1 |

**Findings.**  Using the regression loss *hurts* across all metrics. Therefore, under our setting, **vanilla SFT** is preferable. A plausible cause is that the regression target encourages absolute score mimicry that is misaligned with the iterative tagging objective, which prioritizes well-calibrated, locally contextualized confidence on the current sentence.

## H  DOES OUR TRAINING INDUCE REWARD HACKING?

**Setup.**  We evaluate generations from Llama-3 on the WildHallu benchmark and compare RL-tuned models to non-RL baselines (Vanilla, SFT). We track four simple but sensitive diagnostics:

Table 17: Generation statistics on LLAMA-3 for *WildHallu*. Higher is better for Factuality, Semantic Diversity, and Vocabulary Richness.

| Model | Word Count | Factuality | Semantic Diversity | Vocabulary Richness |
|---|---|---|---|---|
| Vanilla | 134.29 | 0.72 | 0.5688 | 0.5732 |
| SFT | 132.31 | 0.73 | 0.5370 | 0.5738 |
| GRPO | 130.62 | 0.73 | 0.5691 | 0.5670 |
| DPO | 133.50 | 0.74 | 0.5403 | 0.5338 |

- *Word Count* (avg. tokens per output) to detect length hacking.
- *Factuality* (same estimator as in the main results) to ensure truthfulness is not traded away.
- *Semantic Diversity*, computed as the mean embedding cosine *dissimilarity* across outputs:

$$\text{SemDiv} = 1 - \frac{2}{n(n-1)} \sum_{i<j} cos(\mathbf{e}_i, \mathbf{e}_j)$$

  where $\mathbf{e}_i$ is the embedding of the $i$-th sentence within the model's generated paragraph.
- *Vocabulary Richness*, measured by the type–token ratio (TTR): $\text{TTR} = \frac{\#\text{unique tokens}}{\#\text{tokens}}$.

We compute sentence embeddings with `all-MiniLM-L6-v2` using `SentenceTransformers` (Reimers & Gurevych, 2019) and calculate TTR with `nltk`.[1]

**Findings.** RL methods are comparable to baselines on length and factuality and do not reduce semantic diversity or vocabulary richness (Table 17). We additionally audited 100 samples per RL method and found no systematic repetition loops, prompt copying, or template collapse.

Therefore, under our setup, we observe no evidence of reward hacking. While these diagnostics are proxies, they provide a simple, reproducible check that complements the main metrics.

# I NUMERICAL CONSISTENCY

We investigate the probability distribution over decoded confidence tokens to assess whether models have learned to internalize the ordinal structure of the confidence scale. Ideally, a well-calibrated model should rank numerical confidence tokens in an order that reflects their semantic meaning—placing higher probabilities on larger values (*e.g.*, 10 over 9, 9 over 8, etc.) when expressing high certainty.

To probe this behavior, we deliberately select some factually correct statements. We then inspect the top 15 tokens with the highest decoding probabilities. As shown in the following two cases, *all models correctly assign the most probable token as* 10, reflecting high confidence. However, the surrounding distributions reveal key differences.

The GRPO-trained model displays a **near-perfect ordinal alignment**: tokens are ranked in descending order from 10 down to 0, without the presence of extraneous symbols or irrelevant content. This indicates that GRPO not only learns to express high confidence but also internalizes the structure of the confidence scale. In contrast, the DPO model also shows partial ordinal structure, but includes non-numeric or unrelated tokens among its top predictions. We attribute this to DPO's lack of explicit format control, whereas GRPO incorporates a format penalty during training to discourage malformed outputs.

SFT, although it outputs 10 as the most likely token, fails to preserve any consistent ordinal pattern in the rest of the distribution—*e.g.*, lower-confidence values like 0 or 1 may appear above intermediate values. This suggests that SFT does not effectively capture the ordinal relationship between confidence scores, which may contribute to its weaker calibration performance.

More interestingly, such trend holds when the model is uncertain about their output. It demonstrates a desired concave ranking centered at the most probable token. For example in Figure 6, for GRPO,

---

[1]Exact preprocessing: lowercasing, basic punctuation stripping, and whitespace tokenization.

> **Tag on Factually Correct Output**
>
> **Query:**
> In a paragraph, could you tell me what you know about King's College, Cambridge?
> - - - - - - - - - - - - - - - - - - - - - - - - - - - - - - - - - - - - - - - - - - - - - - -
> **Tagging Input:**
> King's College, Cambridge is a constituent college of the University of Cambridge, one of the world's oldest and most prestigious universities. <confidence>

Table 18: Example of model's high confidence output next-token-probability probing.

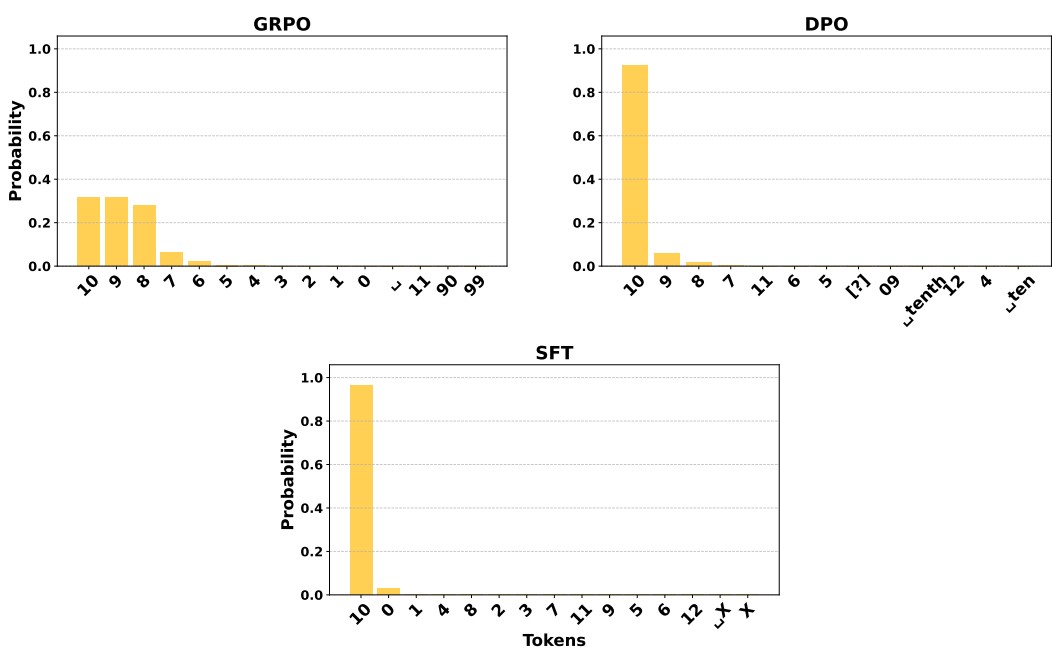

Figure 5: Token probability distributions when the model is confident. The sentence to tag is: "King's College, Cambridge is a constituent college of the University of Cambridge, one of the world's oldest and most prestigious universities."

any confidence score large than it's most probable token 2 is in ascending order, any score smaller in in descending order. Again DPO shows similar pattern but with irrelevant tokens, and SFT fails to grasp the order. These findings reinforce the advantage of reinforcement learning in inducing consistent numerical structure and semantic alignment in confidence estimation.

> **Tag on Factually Incorrect Output**
>
> **Query:**
> In a paragraph, could you tell me what you know about MiniGPT4?
> - - - - - - - - - - - - - - - - - - - - - - - - - - - - - - - - - - - - - - - - - - - - - - -
> **Tagging Input:**
> MiniGPT4 is a lightweight and efficient variant of the popular GPT-4 language model, designed to be more accessible and easier to deploy in resource-constrained environments. <confidence>

Table 19: Example for model's low confidence output for next-token-probability probing.

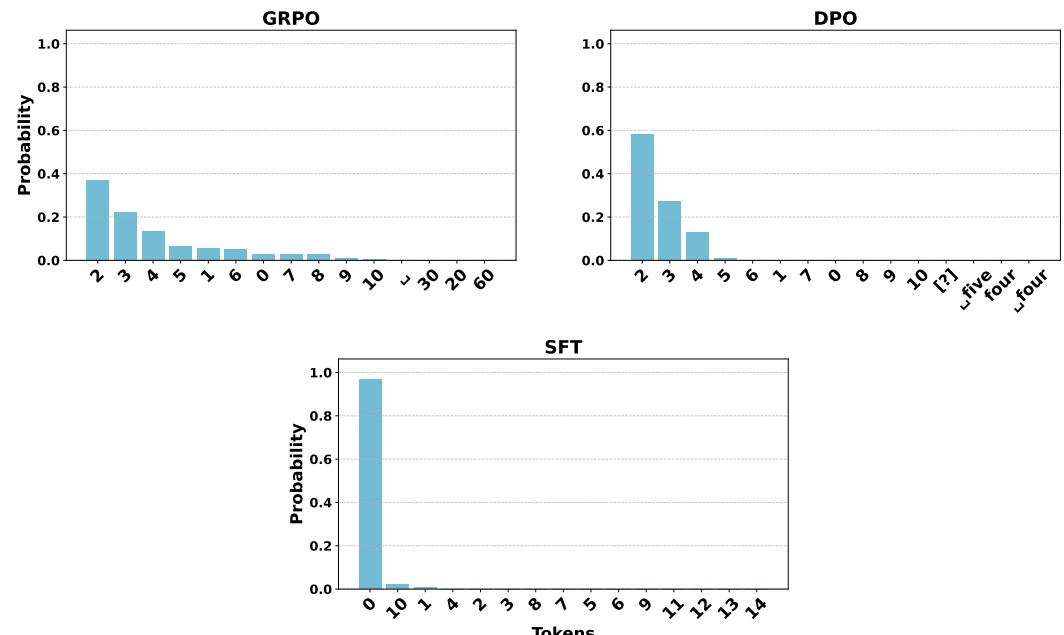

Figure 6: Token probability distributions when the model is uncertain. The sentence to tag is: "MiniGPT4 is a lightweight and efficient variant of the popular GPT-4 language model, designed to be more accessible and easier to deploy in resource-constrained environments."

## J  GENERALIZATION TO SHORT-FORM QA

To assess the generalization capability of our method on short-form confidence estimation, we evaluate it on the TriviaQA Joshi et al. (2017) test set. As illustrated in Table 20, our results show that the proposed method performs on par with sampling-based self-consistency baselines, and closely approaches the performance of the current RL-based state-of-the-art, RewardingDoubt.

Notably, both ours method and RewardingDoubt use the same base model, `Llama-3-8B-Instruct`, and a similar LoRA fine-tuning setup. However, our method is trained on the out-of-domain **Wildhallu** dataset, using only 5.6k examples for a single epoch. In contrast, RewardingDoubt is trained directly on the in-domain TriviaQA dataset, with 174k examples across two epochs. Despite this disparity in domain alignment and data volume, our model achieves a strong approximation to RewardingDoubt's performance.

These results highlight the robustness and domain-transferability of our approach. We believe this test provides encouraging evidence that our method generalizes well to short-form QA tasks, and has the potential for further gains with in-domain fine-tuning.

Table 20: AUROC and ECE metrics for various methods on short-form QA dataset: Trivia QA. All values are presented as percentages.

| Category | Method | AUROC↑ | ECE↓ |
|---|---|---|---|
| Baselines | Self-Verb | 50.0 | 69.3 |
| | p(true) | 60.1 | 21.1 |
| | Self-Cons | 73.4 | 12.2 |
| Literature SOTA | Rewarding Doubt* | 85.9 | 2.2 |
| Our Methods | LoVeC-SFT | 56.3 | 2.0 |
| | LoVeC-GRPO | 69.2 | 6.3 |
| | LoVeC-DPO | 71.2 | 6.9 |

# K  RUNNING TIME

We compare the running time of different confidence estimation methods on **Wildhallu** test set (792 data points) using Gemma2-9B-It and it's RL fine-tuned model on single A100 80GB. Note that Iterative Tagging time counts the generation time of no-confidence facts from the original model. As depicted in 7 below, not only do our methods show better calibration, they also runs $10 \sim 20\times$ faster than sampling based methods, including the literature SOTA, LUQ.

**Why `LoVeC` is faster.**

1. **Single-pass generation.** Self-verification methods (e.g., LUQ, Self-Consistency) resample or decompose outputs and then check them, incurring extra decoding and verifier calls. If $L$ is the number of decoded tokens and $k > 1$ the number of samples, the baseline scales like

$$\underbrace{O(k \cdot L)}_{\text{resampling}} + \underbrace{O(\text{consistency checks})}_{\text{often extra NLI calls}}.$$

LOVEC produces the answer *and* the confidence in the *same* decoding pass:

$$O(L) \quad \text{(no resampling, no separate checker)}.$$

2. **Efficient iterative tagging.** When the passage is fixed, the model emits only a few confidence tokens per sentence, with *no* content re-generation. If $S$ is the number of sentences (and each gets a small constant number of confidence tokens), the per-iteration cost is

$$O(S) \quad \text{(tags only)},$$

whereas self-verification still pays for multiple tag generations and additional checks in each iteration.

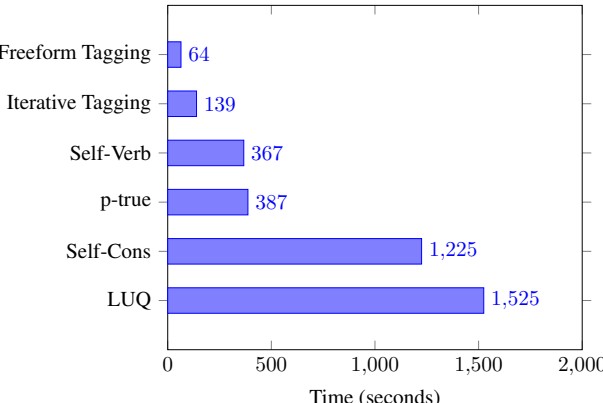

Figure 7: Comparison of total processing time (in seconds) for WildHallu test set (792 samples) using Gemma2-9B-It on single A100 80GB.

## L  CASE STUDY

Table 21: Examples generated by `LoVeC-DPO` from the health domain. The model's low-confidence sentences (highlighted in red) indeed correspond to hallucinated or factually incorrect statements.

---

**Case Study on Health**

**Query:**
In a paragraph, could you tell me what you know about Stiff Person Syndrome?

- - - - - - - - - - - - - - - - - - - - - - - - - - - - - - - - - - - - - - - - - - - - - - - - -

**Model Output:**
Stiff Person Syndrome (SPS) is a rare autoimmune neurological disorder characterized by progressive muscle stiffness and painful spasms, often triggered by stimuli like noise or emotional stress. **<confidence> 8 </confidence>** The condition is caused by a deficiency of dopamine in the brain, leading to impaired muscle coordination. **<confidence> 4 </confidence>** SPS primarily affects the muscles of the trunk and limbs, leading to difficulties in movement and posture. **<confidence> 8 </confidence>** It is commonly treated with antibiotics to eliminate the underlying bacterial infection responsible for the symptoms. **<confidence> 2 </confidence>** Diagnosis often involves detecting antibodies against glutamic acid decarboxylase (GAD) and conducting electromyography (EMG) tests. **<confidence> 7 </confidence>** While there is no cure, treatments like benzodiazepines, baclofen, and immunotherapies can help manage the symptoms effectively. **<confidence> 9 </confidence>**

---

Table 22: Examples generated by `LoVeC-DPO` from the legal domain. The model's low-confidence sentences (highlighted in red) indeed correspond to hallucinated or factually incorrect statements.

---

**Case Study on Law**

**Query:**
In a paragraph, could you tell me what you know about Daubert v. Merrell Dow Pharmaceuticals?

- - - - - - - - - - - - - - - - - - - - - - - - - - - - - - - - - - - - - - - - - - - - - - - - -

**Model Output:**
The 1993 Supreme Court decision in Daubert v. Merrell Dow Pharmaceuticals established the "Daubert standard," which guides federal courts in assessing the admissibility of expert scientific testimony. **<confidence> 9 </confidence>** This standard emphasizes factors such as testability, peer review, error rates, and general acceptance within the relevant scientific community. **<confidence> 10 </confidence>** Under Daubert, the role of evaluating expert evidence shifted from the jury to a specialized panel of scientific advisors appointed by the court. **<confidence> 8 </confidence>** The case arose when plaintiffs alleged that the drug Bendectin caused birth defects, and the Court held that expert testimony must be based on scientifically valid reasoning. **<confidence> 9 </confidence>** Following this ruling, all U.S. states were mandated to adopt the Daubert standard for evaluating expert testimony. **<confidence> 5 </confidence>** The decision underscored the trial judge's role as a "gatekeeper" in ensuring that any and all scientific testimony or evidence admitted is not only relevant but also reliable. **<confidence> 10 </confidence>**

---

# M  PROMPTS

## M.1  PROMPT FOR SFT AND RL-BASED TUNING:

```
You are a helpful assistant. Your task is to provide accurate and
    informative answers to user queries.

For each sentence in your response:
- Include a confidence score from 0, 1, 2, 3, 4, 5, 6, 7, 8, 9, 10.
- The score indicates how certain you are that the sentence is factually
    correct:
    - 0 = very low confidence (likely incorrect)
    - 10 = very high confidence (very likely correct)

Append the confidence score at the end of each sentence using the format:
    <confidence> X </confidence>,
where X is a number from 0 to 10.
```

## M.2  PROMPT FOR P(TRUE):

```
You are required to determine whether a sentence is factually correct
    based on real-world knowledge.
A context is provided for reference, but your judgment should not be
    limited to it.

Respond with 'True' or 'False' only. Do not include any explanations.

Context (for reference): {context}
Sentence: {sentence}

True or False:
```

## M.3  PROMPT FOR VERBALIZED CONFIDENCE (VC):

```
You are required to determine whether a sentence is factually correct
    based on real-world knowledge.
A context is provided for reference, but your judgment should not be
    limited to it.

For each sentence:
- Assign a confidence score from 0, 1, 2, 3, 4, 5, 6, 7, 8, 9, 10.
- The score indicates how certain you are that the sentence is factually
    correct:
    - 0 = very low confidence (likely incorrect)
    - 10 = very high confidence (very likely correct)

Context (for reference): {context}
Sentence: {sentence}

Confidence:
```

## M.4  PROMPT FOR FACT-CHECKING:

```
Your task is to fact-check each of the following sentences.

Each sentence may contain multiple factual claims. For each one:
1. Break down and assess the factual accuracy of each individual detail.
2. Consider names, dates, locations, statistics, events, and attributions
    .
3. Assign a numeric **veracity rating** from 0 to 10 based on this scale:

0: Completely incorrect - entirely false or misleading.
1-3: Mostly incorrect - several key inaccuracies.
4-6: Partially correct - mix of accurate and inaccurate information.
```

```
7-9: Mostly correct - generally accurate with minor issues.
10: Completely correct - all facts are verifiably accurate.

Use the following format for your output (do **not** repeat the sentence)
    :

**Analysis:** [Your detailed factual analysis]
**Rating:** $[0-10]$

---

**Example Inputs:**
### Marie Curie won two Nobel Prizes, one in Physics in 1903 and another
    in Chemistry in 1911 for her work on radioactivity.
### The Great Fire of London occurred in 1666 and destroyed nearly half
    of the city's modern skyscrapers.
### Albert Einstein developed the theory of relativity while working as a
     professor at the University of Zurich and received the Nobel Prize
    in Physics in 1921 for this work.
### Mount Everest, located on the border between Nepal and India, is the
    second-highest mountain in the world after K2.

**Example Outputs:**
**Analysis:** Marie Curie received the Nobel Prize in Physics in 1903 (
    shared with Pierre Curie and Henri Becquerel) and the Nobel Prize in
    Chemistry in 1911 for discovering polonium and radium. The statement
    is entirely accurate.
**Rating:** $10$

**Analysis:** While the date of the fire is correct, the mention of "
    modern skyscrapers" is anachronistic and factually incorrect.
    Skyscrapers did not exist in 1666.
**Rating:** $2$

**Analysis:** Einstein did work at the University of Zurich and received
    the Nobel Prize in 1921, but it was awarded for his explanation of
    the photoelectric effect, not for the theory of relativity.
**Rating:** $6$

**Analysis:** Mount Everest is located between Nepal and the Tibet
    Autonomous Region of China, not India. Additionally, it is the
    highest mountain in the world, not the second-highest.
**Rating:** $1$

---

Here is some relevant information for your reference:

{evidence}

---

Now evaluate the following sentences and output all the results in one go
    .

You should only output the analysis and rating for each sentence without
    repeating the sentences.:

{sentence}
```

