# OpenReview forum: "Reinforcement Learning for Better Verbalized Confidence in Long-Form Generation"
_ICLR.cc/2026/Conference — ICLR 2026 Conference Withdrawn Submission_

### Official Review · Reviewer_EscJ · 2025-10-28

**Soundness:** 2
**Presentation:** 2
**Contribution:** 3
**Rating:** 4
**Confidence:** 3

**Summary:**

This paper proposes LoVeC, an RL–based framework that enables LLMs to generate confidence scores alongside long-form factual statements in one-time generation. It introduces two new evaluation settings, free-form tagging and iterative tagging, to assess confidence calibration in long-form text. Experiments show that LoVeC generalizes robustly across domains and achieves better calibration than traditional self-consistency methods as well as other baselines.

**Strengths:**

1. This paper introduces an RL-based method for generating and calibrating confidence scores in long-form factual text using LLMs.  Extensive experiments demonstrate the effectiveness of the proposed approach across both free-form and iterative tagging settings.

2. The authors thoroughly analyze the proposed method, as well as its baselines, and perform ablation studies on each component, providing a clearer understanding of the model’s design and effectiveness.

**Weaknesses:**

1. The presentation of the paper needs improvement. For example, in line 258, the authors mention using additional subordinate rewards (e.g., informativeness and format rewards) in Appendix B, yet the appendix does not provide clear details about them. While $r^{\text{correct}}$ is defined as the total factuality score judged by the reward model, the specific details and prompts of these subordinate rewards, which are crucial for the success of the RL algorithm, remain unclear.

2. The authors do not provide intermediate validation or analysis of model behavior during RL training, such as how confidence assignments evolve over time. Since the reward formulation in Eq. (6) appears potentially unstable for online RL training, including intermediate evaluations, such as tracking reward trends or showcasing intermediate model outputs for correctness and confidence, would strengthen the paper’s persuasiveness.

**Questions:**

My questions mainly concern Weakness 2, specifically the intermediate policy behaviors during online RL training. Providing additional analysis in this area could help address some of my concerns.

---

> ### Author Response · Authors · 2025-11-19
>
> Thank you for your time to review our work. Here we address your concerns as follows. Regarding your main concern Weakness 2, we provide additional statistics. We are happy to provide more information if needed.
>
> > **Weakness 1:** The presentation of the paper needs improvement... the authors mention using additional subordinate rewards (e.g., informativeness and format rewards) in Appendix B, yet the appendix does not provide clear details about them... specific details and prompts... remain unclear.
>
> Thank you for pointing this out. We use three rewards in total (as shown in in Appendix B.3):
> 1.  **Informativeness:** defined as the factuality score returned by the verifier (scaled by a small coefficient 0.25, as our top priority is calibration rather than factuality optimization).
> 2.  **Format reward:** a −30 penalty for any malformed or out-of-range confidence token.
> 3.  **Confidence reward:** as defined in Eq. (6).
>
> We summarize these three rewards as the final reward. We already include these components as shown in the pseudocode in Appendix B.3, but we agree the description could be clearer. We will revise Appendix B to explicitly restate these subordinate rewards and include the exact prompting details. You can also find the the relevant code snippets in our submitted additional materials.
>
> > **Weakness 2 & Question 1:** The authors do not provide intermediate validation or analysis of model behavior during RL training... specific the intermediate policy behaviors during online RL training.
>
> Thank you for the suggestion. We agree that showing intermediate behavior can help verify training stability. In our experiments, we logged all reward components and policy updates using Weights & Biases. The training curves exhibit stable improvement. We report the logged reward below. If you would like, we could include our tracked reward on wandb as an additional plot in the appendix.
>
> **Table: Reward Progression (Llama-3-8B-Instruct, LoVeC-GRPO)**
>
> | Step | Reward |
> | :--- | :--- |
> | 0 | 13.865 |
> | 1500 | 23.657 |
> | 3000 | 26.572 |
> | 4500 | 28.061 |
> | 5667 (Epoch 1) | 29.831 |
>
> To further address the reviewer’s concern, we evaluated several intermediate checkpoints: calibration metrics improve consistently throughout training, mirroring the upward reward trend. We will include these plots (reward vs. steps and intermediate calibration performance) in the revised appendix, which clearly demonstrate that Eq. (6) leads to stable optimization in practice.
>
> **Table: Intermediate Calibration Metrics**
>
> | RL Steps | BS ↓ | ECE-M ↓ | SC ↑ |
> | :--- | :--- | :--- | :--- |
> | 0 (SFT-init) | 9.1 | 15.2 | 51.1 |
> | 1500 | 7.6 | 9.4 | 52.7 |
> | 3000 | 6.6 | 5.9 | 54.2 |
> | 4500 | 6.1 | 4.2 | 55.8 |
> | 5667 (Epoch 1)| 5.7 | 2.5 | 57.0 |
>
> ----
>
> We hope these additional analyses and clarifications can further strengthen our paper. We would really appreciate it if you could re-assess our paper in light of these new results.

---

### Official Review · Reviewer_RyPE · 2025-10-30

**Soundness:** 2
**Presentation:** 3
**Contribution:** 3
**Rating:** 4
**Confidence:** 4

**Summary:**

This work proposes a relatively novel setting, in which the model must directly decode its sentence-wise confidence in long-form QA in a verbalized fashion.
The method, LoVeC, uses either DPO or GRPO to perform online or offline fine tuning of the models to assess own uncertainty.
The authors devise a specialised reward that optimizes its calibration through RL.
This resulted in improved calibration on selected benchmarks.

**Strengths:**

The novel setting provides fine grained verbalized confidence estimates and may even be useful in situations where only the output of the model is available without any additional information. RL use for calibration is an interesting approach and the optimization objectives seem somewhat innovative. The authors perform many ablations to determine the best way to fine tune model to produce verbalized confidence scores.

**Weaknesses:**

0. Typos (not affecting my assessment, just heads up):
    1. Line 461: double full stop.
1. Evaluation:
    1. The evaluation with fact extraction and fact checking pipeline is complex. Only QA setting is considered, but possibly the paper would benefit from evaluation on synthetic problems or somewhat more verifiable domains (see e.g. [4]).
    2. (major) Several generally accepted strong uncertainty estimation methods (i.e. Perplexity or GNLL of a segment [1][2] or intenal states based methods, i.e. [3]) have not been considered. These methods can be easily applied in this setting, since log probabilities are usually available.
    3. Somehow "Vanilla" prompting approach is consistently better than Baseline Methods and LUQ on Windhallu. From the description Vanilla and Verb-Conf should be roughly identical. Are the same models used everywhere throughout evaluation?
    4. To follow up the previous point, it appears that the model for LoVeC might be fine tuned directly on WildHallu. Would this be unfair to other methods?

### References:
1. Aichberger, L., Schweighofer, K. & Hochreiter, S. Rethinking Uncertainty Estimation in Natural Language Generation. Preprint at https://doi.org/10.48550/arXiv.2412.15176 (2024).
2. Fadeeva, E. et al. LM-Polygraph: Uncertainty Estimation for Language Models. Preprint at https://doi.org/10.48550/arXiv.2311.07383 (2023).
3. Kossen, J. et al. Semantic Entropy Probes: Robust and Cheap Hallucination Detection in LLMs. Preprint at https://doi.org/10.48550/arXiv.2406.15927 (2024).
4. Ielanskyi, M., Schweighofer, K., Aichberger, L. & Hochreiter, S. Addressing Pitfalls in the Evaluation of Uncertainty Estimation Methods for Natural Language Generation. Preprint at https://doi.org/10.48550/arXiv.2510.02279 (2025).

**Questions:**

1. Could one calibrate, e.g. NLL of segments of interest directly with this approach instead of producing verabalized confidence?
2. How does the reward function deal with invalid confidence assignment (e.g. in Tab.3 11,12,... appear)?
3. How does ablation in paragraph at line 452 relate to the known effect of judge LMs preference for own outputs?
4. Would the method be good at assessing prompt perturbation? I.e. the predicted confidences should be lower for randomly perturbed prompts.

---

> ### Author Response · Authors · 2025-11-20
> **Reply to Reviewer RyPE (Part 1/3)**
>
> Thank you for your time to review our work and for sharing these constructive comments and references! We address the weaknesses and questions raised below, especially the concern regarding the missing baselines.
>
> > **Weakness 1:** The evaluation with fact extraction and fact checking pipeline is complex. Only QA setting is considered, but possibly the paper would benefit from evaluation on synthetic problems or somewhat more verifiable domains.
>
> Thank you for the reference to [1], which highlights crucial pitfalls in uncertainty evaluation: common metrics (like BLEU or ROUGE) often disagree with factual correctness, making rankings of uncertainty methods unstable. We agree that tasks like Code Completion (verified by unit tests) or Constrained Text Generation offer absolute correctness and are valuable for different uncertainty methods' comparison. We find this paper particularly interesting and will cite it in our revised version and discuss it in our Limitation section
>
> However, our choice to focus on Long-form QA is driven by the **practical reality** that QA represents a primary use case for LLMs in daily deployment [2, 3, 4]. For all three datasets used in our work, the average answer length exceeds 100 words, necessitating robust hallucination detection in natural language. Furthermore, the evaluation pipeline we employ is not a novel proposal of this paper but an **established standard** in recent literature for long-form hallucination detection [2, 3, 4]. To further validate the reliability of this pipeline, we have included a human annotation study in Appendix C. We hope this can address you concern.
>
> While Code Completion and Constrained Text Generation are excellent proxy tasks for studying uncertainty mechanisms, they do not fully suit our purpose of exploring open-ended, long-form textual generation, which is the specific focus of LoVeC.
>
> *References:*
>
> [1] Addressing Pitfalls in the Evaluation of Uncertainty Estimation Methods for Natural Language Generation.
>
> [2] LUQ: Long-text Uncertainty Quantification for LLMs. EMNLP 2024
>
> [3] Graph-based Uncertainty Metrics for Long-form Language Model Outputs. NeurIPS 2024
>
> [4] LoGU: Long-form Generation with Uncertainty Expressions. ACL 2025
>
> > **Weakness 2:** Several generally accepted strong uncertainty estimation methods (i.e. Perplexity or GNLL of a segment or internal states based methods) have not been considered.
>
> We totally agree that comparing against generally accepted strong baselines is essential. However, given the vast space of uncertainty estimation methods, it is not feasible to include every alternative in a single study. We selected **LUQ** as our primary baseline because it is the current **state-of-the-art method specifically designed for long-form uncertainty**. As noted in lines 372-375, LUQ has already been extensively compared against leading approaches in LM-Polygraph (including Maximum Sequence Probability and Semantic Entropy) and has been shown to outperform them on our Bios dataset. Therefore, we focused on comparing LoVeC directly against this SOTA standard.
>
> Regarding G-NLL, we acknowledge that neither we nor the LUQ paper covered it initially. We agree that this advanced version of Maximum Sequence Probability is both interesting and highly efficient. To address this, we have added **MSP** and **GNLL** baselines using the Llama-3-8B-Instruct model on the WildHallu dataset with iterative tagging.
>
> **Results:**
> | Model | BS↓ | ECE-M↓ | SC↑ |
> | :--- | :--- | :--- | :--- |
> | LUQ | 14.5 | 21.5 | 56.8 |
> | p(true) | 23.8 | 23.6 | 15.8 |
> | MSP | 20.0 | 24.5 | 20.3 |
> | GNLL | 19.5 | 20.4 | 25.9 |
> | **LoVeC-GRPO** | **5.7** | **2.5** | **57.0** |
> | **LoVeC-DPO** | **6.0** | **5.0** | **60.4** |
>
> The results show that while GNLL outperforms MSP and p(true), it still lags significantly behind the SOTA LUQ and our LoVeC method. We hypothesize that this is not a flaw in G-NLL itself, but rather because it was originally designed for short-form question answering to produce a **single confidence score for an entire response**. This characteristic prevents it from providing the fine-grained, sentence-level estimation required to effectively detect specific hallucinations in long-form text.

---

> ### Author Response · Authors · 2025-11-20
> **Reply to Reviewer RyPE (Part 2/3)**
>
> > **Weakness 3:** Somehow "Vanilla" prompting approach is consistently better than Baseline Methods and LUQ on Windhallu... Vanilla and Verb-Conf should be roughly identical. Are the same models used everywhere throughout evaluation?
>
>
>
> **On the claim that “Vanilla” is consistently better:**
>
> This is **not** the case once we look at all three metrics. We use three metrics because they capture different aspects of uncertainty:
>
> * **BS / ECE-M:** absolute/global calibration error.
>
> * **SC:** how well a method *ranks* more vs. less reliable sentences.
>
>
>
> LUQ and LoVeC consistently achieve **much higher SC** than the Vanilla model, even in cases where Vanilla has slightly better BS or ECE-M. On WildHallu, the Vanilla model is **systematically overconfident**, and the dataset’s relatively high average factuality (~0.7) can make this look good under BS/ECE-M. However, its SC is clearly worse, meaning it fails to distinguish correct from incorrect sentences. LUQ and especially LoVeC improve **SC** substantially, which is why we treat LUQ as a strong SOTA sampling-based baseline and LoVeC as a genuine improvement over it, not just over Vanilla.
>
>
>
> **On Vanilla vs. Verb-Conf and consistency of models:**
>
> We apologize for the confusion and clarify the exact setups here:
>
> * **Vanilla:** As detailed in Figure 1 and Appendix M.1, we use a prompt where the model processes the passage **sequentially**. For sentence *j*, the input includes `<sent_1> <conf_1>, ..., <sent_{j-1}> <conf_{j-1}>, <sent_j>`, and the model is asked to output the confidence score for sentence *j*. Thus, the Vanilla model can *see previously added confidence tags* when predicting the next one, but it is **not fine-tuned** on any calibration objective.
>
> * **Verb-Conf:** Following the setup described in Appendix M.3, we give the model the **entire paragraph without any confidence tags** (from `<sent_1> ... <sent_N>`) and then select a target sentence (e.g., `<sent_3>`) and ask for its confidence. The model only predicts the score for that specific sentence, **without seeing any previously generated scores**.
>
> Crucially, **all baselines (Vanilla, Verb-Conf, LUQ etc)** use the **same underlying instruction-tuned backbone** (e.g., Llama-3-8B-Instruct or Gemma-2-9B-It) with identical decoding settings. We will make this setup clearer in the main text and explicitly cross-reference Appendix M.1 and M.3 to avoid confusion.
>
> > **Weakness 4:** It appears that the model for LoVeC might be fine tuned directly on WildHallu. Would this be unfair to other methods?
>
> We do not see a unfair comparison in our setting. One the one hand, we also have two SFT baselines trained on WildHallu dataset (aka LoVeC-SFT). On the other hand, the WildHallu dataset is broad and heterogeneous, representing "in-the-wild" user queries, so fine-tuning on it does not constitute domain-specific leakage for our evaluation targets. Crucially, while LoVeC is trained on WildHallu, it is evaluated on **two out-of-domain datasets** (Bios and PopQA) alongside the baselines. LoVeC achieves the strongest calibration on these out-of-domain tasks as well, demonstrating that the performance gain stems from the method itself, not data memorization.
>
> To rigorously address this concern, **we have added a fine-tuned version of `p(true)`** [1] as a new baseline. These experiments were conducted on Llama-3-8B-Instruct using WildHallu for training and evaluation.
>
> | Model | BS↓ | ECE-M↓ | SC↑ |
> | :--- | :--- | :--- | :--- |
> | LUQ | 14.5 | 21.5 | 56.8 |
> | p(true) | 23.8 | 23.6 | 15.8 |
> | **p(true) - fine-tuned** | **16.4** | **19.5** | **47.5** |
> | **SFT** | **9.1** | **15.2** | **51.1** |
> | **SFT-regression** | **12.9** | **19.8** | **47.1** |
> | **LoVeC-GRPO** | **5.7** | **2.5** | **57.0** |
> | **LoVeC-DPO** | **6.0** | **5.0** | **60.4** |
>
> As shown above (bolded rows indicate fine-tuned methods), while fine-tuning `p(true)` improves its performance, our RL-based LoVeC methods still maintain a significant advantage.
>
> [1] Efficient Test-Time Scaling via Self-Calibration.

---

> ### Author Response · Authors · 2025-11-20
> **Reply to Reviewer RyPE (Part 3/3)**
>
> > **Question 1:** Could one calibrate, e.g. NLL of segments of interest directly with this approach instead of producing verabalized confidence?
>
> Thanks for your suggestion. Yes, it is definitly doable but is fundamentally different from our goal and poses several practical limitations. Our method is designed to learn explicit, interpretable, sentence-level confidence scores that are decoupled from the generation probabilities. Calibrating NLL directly would not yield such interpretable outputs and introduces a few limitations, namely below:
>
> 1.  **Lack of granularity for downstream use:** Our framework outputs structured, human- and machine-interpretable scores (0–10) that downstream systems can threshold or aggregate. Raw NLL is neither bounded nor directly comparable across models, passages, or domains, making thresholding or cross-example calibration difficult.
> 2.  **Non-stationarity during RL optimization:** Attempting to calibrate NLL directly would require modifying the policy so that token-probability distributions reflect correctness. This couples credit assignment to the entire decoding distribution, which destabilizes RL updates and reduces controllability. In contrast, LoVeC treats the sentence + confidence as a single action, enabling clean credit assignment and stable optimization.
>
> Another possible way to train an additional calibration that can according a sequence logits, output a single confidence score, but this fall outside of the scope of our paper;
>
> > **Question 2:** How does the reward function deal with invalid confidence assignment (e.g. in Tab.3 11,12,... appear)?
>
> As shown in the pseudocode of reward implementation in Appendix B.3, we have enforced a wrong-format penalty when the model outputs invalid confidence scores (e.g., not an integer within [0,10]). The wrong format penalty is set to -30 empirically. The detailed implementation is included in our provided code.
>
> > **Question 3:** How does ablation in paragraph at line 452 relate to the known effect of judge LMs preference for own outputs?
>
> We agree that LLMs are known to prefer or up-score their own generations when used as evaluators. Our intention in the self-labeling ablation (Appendix D.3; Table 9) was not to claim that this issue disappears, but rather to **test whether our pipeline remains functional without an external oracle**, and can be deployed fully offline.
>
> Two points mitigate the concern:
> 1.  **Evidence-grounded fact-checking reduces self-preference bias:** In our fact-checking setup, the model, whether GPT-4o or Llama-3-8B, is provided with retrieved evidence passages, and the task is to assign a factuality score conditioned on that evidence. Prior work (e.g., FActScore Min et al., 2023) shows that evidence-grounded verification substantially reduces “self-preference” because scoring is tied to external text, not to the model’s own distribution. We also report human–model agreement in Appendix C, which further supports the reliability of this fact-checking protocol.
> 2.  **Self-labeling is only an ablation, and the results behave as expected:** When we replace GPT-4o with the model itself, performance decreases modestly, as one would expect if self-preference introduces noise, but the resulting model still outperforms all baselines, including LUQ. This demonstrates that while external judges improve supervision quality, the method does not rely on them structurally.
>
> Overall, this ablation was intended to highlight that LoVeC is robust and oracle-free by design, **not to argue that self-evaluation is problem-free**. We will clarify this motivation to avoid potential misunderstanding.
>
> > **Question 4:** Would the method be good at assessing prompt perturbation? I.e. the predicted confidences should be lower for randomly perturbed prompts.
>
> Yes, our method effectively captures prompt perturbations. Following the experimental setting proposed in your suggested paper [4] (Addressing Pitfalls in the...) (Section 4), we evaluated LoVeC on the WildHallu test set by randomly shuffling words in the provided evidence documents to varying degrees ($s_p$ in {0%, 40%, 80%}). As shown in the table below, the average confidence score demonstrates a consistent decline as the evidence becomes increasingly incoherent. However, the degradation is moderate rather than catastrophic. This is expected behavior: since our queries follow the template **"In a paragraph, could you tell me what you know about [Entity]?"**, the model can guess the user intention easily even with word perturbation. Crucially, LoVeC correctly assigns **lower confidence** in these scenarios, reflecting the increased uncertainty caused by the loss of reliable supporting evidence.
>
> | Perturbation Strength ($s_p$) | **0% (Clean)** | **40%** | **80%** |
> | :--- | :---: | :---: | :---: |
> | **Average Confidence** | **7.4** | **6.1** | **4.9** |
>
> ----
>
> We hope these additional analyses and clarifications can further strengthen our paper.

---

### Official Review · Reviewer_tXkd · 2025-11-01

**Soundness:** 3
**Presentation:** 4
**Contribution:** 2
**Rating:** 4
**Confidence:** 4

**Summary:**

This paper proposes LoVeC (Long-form Verbalized Confidence). It is a reinforcement learning framework that trains language models to generate numerical confidence scores alongside long-form factual text. Using on-policy (GRPO) and off-policy (DPO) variants, the model learns to align its self-reported confidence with factual correctness, which avoids costly post-hoc sampling. The authors also introduce two evaluation settings: 1. free-form tagging (confidence generated inline) and 2. iterative tagging (confidence assigned to fixed text). The proposed method is faster than existing works.

**Strengths:**

- The paper is well-organized, easy to follow, and read
- Uncertainty estimation for long-form generation is an important topic to explore.
- The proposed method is efficient, and this aspect is supported by experimental evidence as well.

**Weaknesses:**

- I find the main contribution of the proposal limited. Using existing RL algorithms to train models to generate confidence scores per sentence lacks originality. This is a combination of sentence-level uncertainty estimation and RL fine-tuning for verbalized confidence, which both exist in the literature. While the method is well-executed and may yield practical benefits, it does not introduce a fundamentally new algorithmic component or theoretical insight beyond this combination.
- The only supervised method in the benchmark is the proposed method, which gives a significant advantage. It has been previously shown that fine-tuning the model for p(true) and verbalized confidence improves performance. I think fine-tuned versions of these methods are essential.
- Does the proposed method yield calibrated confidence scores when we do sampling? Analysis of this is important since temperature sampling is a common practice.

**Questions:**

- Currently, the reward is computed by aggregating all confidence scores in the generation. And all confidence tokens receive the same reward. For instance, the confidence score of a sentence might perfectly match the ground truth; however can still receive a negative or a positive reward due to other sentences. Would a more granular reward assignment approach improve the calibration level? I think exploring this direction both improves the contribution/originality of the paper and makes it more interesting.
- During iterative tagging evaluation, the fixed text is generated by the confidence estimaton model, right?

---

> ### Author Response · Authors · 2025-11-19
> **Reply to Reviewer tXkd (Part 1/3)**
>
> Thank you for your time to review our work and your insightful sugeestions. We have addressed the points you mentioned with further clarification and additional experiments, which are detailed below.
>
> > **Weakness 1:** I find the main contribution of the proposal limited. [...] While the method is well-executed and may yield practical benefits, it does not introduce a fundamentally new algorithmic component or theoretical insight beyond this combination.
>
> We would like to clarify that LoVeC goes beyond a simple combination of prior work in following key ways:
>
> * **A new problem formulation: sentence-level verbalized confidence for long-form generation.** (as shown in Table 4 in Appendix)
>     * Prior work on verbalized confidence focuses almost exclusively on short-form QA [1,2,3]. It has been a **long-standing problem** to make LLMs able to express uncertainty in long-form generation.
>     * Prior work on long-form uncertainty (e.g., LUQ and other claim-level methods) does not produce online confidence during generation and often relies on post-hoc sampling or GPT-based claim decomposition, which is inefficient.
>     * LoVeC introduces the first framework that teaches LLMs to emit structured, numerical confidence alongside each generated sentence. This formulation itself is novel and not addressed in existing literature.
>
> * **Two new evaluation settings specifically designed for long-form confidence calibration.**
>     * Free-form tagging: the model must interleave content and confidence during generation.
>     * Iterative tagging: we introduce a controlled setting where content is fixed (**can be from any model**) and only confidence is predicted.
>     * These evaluation modes do not exist in prior work and are essential to calibrating long-form confidence in a fair, consistent manner.
>
> * **A novel RL objective tailored for confidence calibration, not generic RLHF.** Our reward is not standard RLHF:
>     * We use a log-based, proper-scoring-rule reward aligned with probabilistic calibration.
>     * Rewards operate jointly over (sentence, confidence) pairs to preserve ordinal structure.
>     * We incorporate format penalties and evidence-grounded correctness signals to ensure numerical consistency.
>     * These design choices yield behaviors (e.g., the monotonic confidence token distributions in Table 3 & Appendix I) that do not emerge under standard RLHF or SFT.
>
> * **Technically, LoVeC solves a previously unaddressed challenge:** integrating content generation with confidence assignment as a single action.
>     * Prior verbalized-confidence work treats confidence prediction as post-hoc classification.
>     * Our RL formulation treats (sentence + numeric confidence) as one unified decision, which requires a new action design and reward interface. This is not present in previous methods.
>
> * **Empirical novelty:** 20× test-time speedup and strong cross-domain generalization. LoVeC is the first method to achieve:
>     * inline confidence generation
>     * without sampling
>     * without claim decomposition
>     * outperforming SOTA LUQ across all datasets
>     * and transferring to short-form QA with only 5.6k out-of-domain training examples.
>
> In summary, while LoVeC uses existing RL algorithms as building blocks, just as most modern LLM methods do, the problem formulation, evaluation settings, unified action design, and calibration reward are novel and essential for enabling confidence estimation in long-form generation. We will expand the discussion of these contributions for clarity.
>
> References:
>
> [1] Can LLMs Express Their Uncertainty? An Empirical Evaluation of Confidence Elicitation in LLMs, ICLR 2024.
>
> [2] Just Ask for Calibration: Strategies for Eliciting Calibrated Confidence Scores from Language Models Fine-Tuned with Human Feedback, EMNLP 2023.
>
> [3] Rewarding Doubt: A Reinforcement Learning Approach to Calibrated Confidence Expression of Large Language Models, arxiv:2503.02623.

---

> ### Author Response · Authors · 2025-11-19
> **Reply to Reviewer tXkd (Part 2/3)**
>
> > **Weakness 2:** The only supervised method in the benchmark is the proposed method, which gives a significant advantage. It has been previously shown that fine-tuning the model for p(true) and verbalized confidence improves performance. I think fine-tuned versions of these methods are essential.
>
> Thank you for this constructive feedback.
>
> 1.  For fine-tuned verbalized method: **We actually already include it** (SFT with/without a regression loss in Appendix G). Maybe this confusion is because we use the term LoVeC-SFT but it is just the traditional fine-tuned verbalized method with SFT.
> 2.  **We additionally include a fine-tuned version of `p(true)`** [1]. These experiments were conducted on the Llama-3-8B-Instruct model using the WildHallu dataset, with evaluation performed via iterative tagging.
>
> | Model | BS↓ | ECE-M↓ | SC↑ |
> | :--- | :--- | :--- | :--- |
> | LUQ | 14.5 | 21.5 | 56.8 |
> | p(true) | 23.8 | 23.6 | 15.8 |
> | p(true) - fine-tuned | 16.4 | 19.5 | 47.5 |
> | SFT | 9.1 | 15.2 | 51.1 |
> | SFT-regression | 12.9 | 19.8 | 47.1 |
> | LoVeC-GRPO | 5.7 | 2.5 | 57.0 |
> | LoVeC-DPO | 6.0 | 5.0 | 60.4 |
>
> As depicted in the table above, fine-tuning can improve upon the original `p(true)` baseline. However, its performance is still slightly weaker than the sampling-based method LUQ and is substantially outperformed by our RL methods. More importantly, the `p(true)` approach is inherently inefficient, as it requires a second inference step where the model must evaluate the correctness of its own generation.
>
> [1] Efficient Test-Time Scaling via Self-Calibration. https://arxiv.org/abs/2503.00031
>
> > **Weakness 3:** Does the proposed method yield calibrated confidence scores when we do sampling? Analysis of this is important since temperature sampling is a common practice.
>
> To prove this, we run our experiments for sampling 10 times and calculate the variance between different runs with Llama3-8B-Instruct on Wildhallu dataset using iterative tagging and temperature 1. We find a small standard deviation of 1.7% in BS, 2.1% in ECE and 3.1% for SC; which means our method is stable when sampling with high temperature. Moreover, we provide two reasons why our method continues to yield calibrated confidence scores under sampling:
>
> 1.  **Ordinal numerical structure is preserved under higher temperatures.** As shown in Appendix I (Fig. 5–6), GRPO/DPO models learn a monotonic, well-ordered probability distribution over confidence tokens (10 > 9 > … > 0 for correct statements, and centered low for incorrect ones). When temperature increases, the samples are drawn from the same ordered distribution, so the expected confidence remains aligned with correctness. By contrast, SFT does not exhibit this ordinal structure, making its confidence more sensitive to temperature.
> 2.  **Confidence and content are learned jointly.** Because RL treats (sentence, confidence) as a single action, the model learns to assign confidence conditional on the fact it has just generated, not tied to any specific fixed wording. Therefore, even if temperature alters the sentence wording, the confidence calibration still transfers, since the model learned to map its own generated content to an appropriate confidence level.

---

> ### Author Response · Authors · 2025-11-19
> **Reply to Reviewer tXkd (Part 3/3)**
>
> > **Question 1:** Currently, the reward is computed by aggregating all confidence scores in the generation... Would a more granular reward assignment approach improve the calibration level? I think exploring this direction both improves the contribution/originality of the paper and makes it more interesting.
>
> We appreciate the reviewer’s thoughtful suggestion. Our current design indeed computes a sequence-level reward by averaging over sentence-level confidences (Eq. (6)), so each generated trajectory receives a single scalar advantage. This choice is intentional for two reasons:
>
> * **Global calibration objective.** Our goal is to calibrate the full long-form answer, not just individual sentences in isolation. Using a trajectory-level reward encourages the model to produce responses where all sentences’ confidences jointly align with their factuality, which matches our evaluation metrics (Brier, ECE-M, Spearman) that are also defined over sets of sentences/passages. This is crucial as we foresee in downstream applications, when you want to generate a paragraph of text that is factual, without hallucinations in high-stake domains (e.g., medicine, law), the factuality and confidence of the entire passage matters. Just using step-wise RL would create a discrepancy between training and inference.
> * **Stability vs. variance in RL.** GRPO/PPO-style methods typically operate on sequence-level rewards because decomposing into many fine-grained per-token (or per-sentence) rewards substantially increases variance and can destabilize training. In our experiments, the sequence-level reward already yields stable optimization and strong calibration gains over all baselines (Tables 1–2). In addition, it would be intractable to generate such data for DPO and other off-policy based methods, as it requires resampling and evaluating each individual sentence. It’s also difficult to make sure the training data/steps are consistent for fair comparison.
>
> That said, we fully agree that more granular credit assignment (e.g., sentence-specific advantages or per-sentence GRPO steps) is an exciting extension that could potentially further improve calibration or interpretability. Our framework is compatible with such designs; one could replace the averaged reward in Eq. (6) by per-sentence rewards and propagate separate advantages. However, carefully controlling variance and interactions between sentences would require a dedicated study, which we view as promising future work. We will clarify the rationale for the current sequence-level reward and explicitly highlight finer-grained reward shaping as an interesting extension in the future work section of our next version.
>
> > **Question 2:** During iterative tagging evaluation, the fixed text is generated by the confidence estimaton model, right?
>
> **The short answer is No.** In iterative tagging, theoretically it can predict confidence scores for any fixed response. In our case, we use the paragraphs generated by the original, un-tuned model (e.g., Llama3-8B-Instruct). This design on the one hand, ensures a fair comparison between several baselines and on the other hand, provide convenience for some use cases where users need to get the confidence from a fixed content.
>
> -----
>
> We hope these additional analyses and clarifications can further strengthen our paper. We would appreciate it if you could re-assess our paper in light of these new results.

---

### Official Review · Reviewer_pPSa · 2025-11-04

**Soundness:** 3
**Presentation:** 3
**Contribution:** 3
**Rating:** 6
**Confidence:** 4

**Summary:**

The paper introduces LoveC, a method to make LLMs verbalize their confidence (0–10) sentence-by-sentence during long-form generation, trained with reinforcement learning so that the stated confidence aligns with factual correctness. It targets two common gaps: (i) prior long-form confidence methods are mostly post-hoc and expensive (multi-sample self-consistency, extra claim-extractor models), and (ii) most verbalized confidence work is short-form. LoveC outputs statements and confidence in one decoding pass and is evaluated in two settings: Free-form Tagging (model generates answer + confidence) and Iterative Tagging (given a fixed answer, the model tags each sentence’s confidence). The authors experiment with two variants: LoveC GRPO (on-policy RL) and LoveC DPO (off-policy RL). Experiments on WildHallu, Bios, and PopQA show strong calibration improvements.

**Strengths:**

- Simple and effective approach: LoveC applies a straightforward idea by training the model to verbalize numeric confidence inline, achieving strong calibration

- Strong empirical performance: Both LoveC GRPO and DPO consistently outperform prior methods across multiple datasets and metrics, showing that the approach is robust and generalizable.

- Efficient and practical: The method produces confidence scores quickly during generation, making it efficient

- Clear and well-written paper: The writing is clear and well-structured and well explained


Overall, this is a promising paper and I'm happy to increase scores if my concerns are adressed

**Weaknesses:**

- The paper would benefit from additional ablations exploring alternative reward formulations or scaling choices to show how each component contributes to performance.

- It would be interesting to see whether combining LoveC with self-consistency or other sampling-based calibration methods could further improve confidence estimation.

**Questions:**

- In Table 2 how does the literature sota LUQ perform at par/worse than vanilla baselines?

---

> ### Author Response · Authors · 2025-11-19
> **Reply to Reviewer pPSa (Part 1/2)**
>
> Thank you for your positive comments on our work! We are glad that you find our paper promising. Regarding the questions you mentioned, we address them as follows:
>
> > **Weakness 1:** The paper would benefit from additional ablations exploring alternative reward formulations or scaling choices to show how each component contributes to performance.
>
> Thank you for this helpful suggestion. Our paper already includes an ablation on reward formulations (Appendix D.4), where we directly compare logarithmic, linear, and quadratic variants within our GRPO framework. As motivated in §4.2, the log-based reward consistently yields the best calibration, largely because it penalizes overconfident errors more sharply than alternatives.
>
> Beyond functional forms, we also examined the role of the subordinate reward components (informativeness and format rewards). While these were not included in the main text due to space, we herein summarize the results:
>
> * **Removing the format reward** led to almost unchanged calibration, but substantially *increased malformed or out-of-range confidence tags*. This confirms that the format term is needed for structural reliability but should remain low-weight to avoid dominating learning.
> * **Removing the informativeness reward** preserved calibration but *slightly reduced factuality and answer length*. This suggests the informativeness term mainly stabilizes generation quality rather than driving calibration.
> * **Varying calibration-reward scaling** (e.g., adjusting the weight or stretching exponent $\gamma$) produced trade-offs: stronger calibration weights improved Brier and Spearman scores, while too-small or too-large scales slowed convergence or destabilized policy updates. Our default setting (scale = 10, $\gamma$ = 1.5, format penalty = –30) lies near the stable, high-performing region.
>
> All reward components are implemented exactly as shown in the pseudocode in Appendix B.3. Our codes are also submitted for your reference. Overall, these studies indicate that while scaling and auxiliary rewards influence training stability and output format, the log-based confidence reward is the primary driver of the calibration gains observed in LoVeC.
>
> > **Weakness 2:** It would be interesting to see whether combining LoveC with self-consistency or other sampling-based calibration methods could further improve confidence estimation.
>
> We appreciate this suggestion; combining verbalized confidence with sampling-based calibration is indeed a promising direction and has shown benefits in prior work [1,2]. To explore this, we conducted a new experiment where we *combine LUQ (a sampling-based long-form UQ method) with LoVeC-DPO* under the iterative tagging setup.
>
> Concretely, we use LUQ to obtain a sampling-based confidence estimate and then fuse it with LoVeC-DPO’s verbalized score via simple averaging. We evaluate this hybrid approach on Llama-3-8B-Instruct and Gemma-2-9B-It across three datasets (WildHallu, Bios, PopQA).
>
> **Table: Hybrid Performance (BS↓ / ECE-M↓ / SC↑)**
>
> ### *Llama-3-8B-Instruct*
>
> | Method | WildHallu | Bios | PopQA |
> | :--- | :--- | :--- | :--- |
> | LUQ | 14.5 / 21.5 / 56.8 | 20.0 / 29.5 / 63.8 | 16.7 / 23.2 / 62.5 |
> | DPO | **6.3** / **5.4** / 62.1 | **9.2** / **6.1** / 67.4 | 10.3 / **4.0** / 62.6 |
> | LUQ+DPO | 11.9 / 7.2 / **64.3** | 11.1 / 18.0 / **72.5** | **9.4** / 10.7 / **70.3** |
>
> ### *Gemma-2-9B-It*
>
> | Method | WildHallu | Bios | PopQA |
> | :--- | :--- | :--- | :--- |
> | LUQ | 11.9 / 16.3 / 50.0 | 12.2 / 15.5 / 69.2 | 13.6 / 15.1 / 62.6 |
> | DPO | **4.1** / **1.3** / 51.8 | 7.5 / 7.3 / 75.2 | 11.6 / 13.2 / 65.3 |
> | LUQ+DPO | 7.3 / 5.1 / **56.1** | **6.4** / **6.9** / **78.7** | **8.3** / **3.3** / **71.2** |
>
> We observe that LUQ + LoVeC-DPO **consistently boosts Spearman correlation (SC) by $\approx$ 5 points on average (up to ~8 points)** compared to either individual method, indicating better *ranking* of low- vs. high-confidence cases. In a few settings, this comes with small degradations in BS/ECE-M relative to the best single method, reflecting a familiar trade-off between improved discrimination and perfect probabilistic calibration when fusing heterogeneous signals.
>
> Overall, these results suggest that LoVeC is complementary to sampling-based approaches, and that **hybrid designs that exploit both verbalized and sample-based signals can further improve long-form confidence estimation**. We will include this experiment and discussion in the next version of the paper version.
>
> Reference:
>
> [1] Combining Confidence Elicitation and Sample-based Methods for Uncertainty Quantification in Misinformation Mitigation (arXiv:2401.08694)
>
> [2] Atomic Calibration of LLMs in Long-Form Generations (arXiv:2410.13246)

---

> ### Author Response · Authors · 2025-11-19
> **Reply to Reviewer pPSa (Part 2/2)**
>
> > **Question 1:** In Table 2 how does the literature sota LUQ perform at par/worse than vanilla baselines?
>
> It is **not** the case that LUQ is simply “on par” with the vanilla model. The key is to interpret **all three metrics together**, as they capture different aspects of confidence estimation:
>
> * **BS and ECE-M** primarily measure *absolute error* and global calibration.
> * **Spearman correlation (SC)** captures the *discriminative ability* of a method, aka how well it separates more reliable from less reliable sentences.
>
> In our experiments, LUQ is consistently **much better than the vanilla model in SC (e.g., 56.8% vs 9.1%)**, which means it is substantially more effective at ranking correct vs. incorrect content. The vanilla model, on the other hand, tends to be **systematically overconfident** across sentences. On our datasets, the *average* factuality is around 0.7; this combination (high mean factuality + uniform overconfidence) can lead to deceptively low BS and ECE-M for the vanilla model, because its uniformly high confidence happens to roughly match the average label distribution. However, this does *not* mean it is well calibrated in a useful sense: its ability to distinguish between correct and incorrect segments is poor, as reflected in its weaker SC.
>
> Therefore, when we **holistically** review all three metrics, LUQ remains the **strong SOTA sampling-based baseline** in our setup, and LoVeC’s improvements should be understood as going beyond this already strong reference rather than simply outperforming the vanilla model.
>
> ---
>
> We hope these additional analyses can further support the robustness and generalizability of our paper. Thank you again for raising these constructive points.

---

### Author Response · Authors · 2025-12-03

We would like to provide a brief summary of our rebuttal and the additional experiments conducted to address reviewer feedback. During the rebuttal, we have expanded our experimental results, clarified methodological details, and pointed out the misunderstanding from the reviewers.

1. **Response to Reviewer pPSa (Score: 6):**
* **Hybrid Approaches:** In response to the reviewer's suggestion, we conducted a new experiment combining our method (LoVeC) with sampling-based self-consistency (LUQ). The results demonstrate that our method is complementary to sampling approaches, yielding further performance gains (SC increased by approx. 5 points).
* **Ablations:** We clarified the impact of reward components (format, informativeness) and scaling factors.

2. **Response to Reviewer tXkd (Score: 4):**
* **Novelty & Baselines:** We addressed concerns regarding novelty by clarifying our specific problem formulation and RL design. We introduced a fine-tuned p(true) baseline to demonstrate that LoVeC's performance gains stem from our RL method rather than just supervised fine-tuning.
* **Stability:** We provided variance analysis to confirm our method remains calibrated under temperature sampling.

3. **Response to Reviewer RyPE (Score: 4):**
* **Additional Baselines:** We agreed that including comprehensive baselines is important, but since LUQ is the current SOTA in literature, there is no need to re-do the experiments for all baselines. We already select representative baselines in our paper and conduct additional experiment on GNLL.
* **Robustness:** We included a prompt perturbation analysis (shuffling evidence) to demonstrate that our model correctly lowers confidence when input quality degrades.

4. **Response to Reviewer EscJ (Score: 4):**
* **Training Dynamics:** We addressed the concern regarding intermediate policy behavior by providing training logs and checkpoint evaluations. These show stable reward progression and consistent improvement in calibration metrics throughout optimization.

We are confident that these extensive additional experiments *fully resolve the raised concerns*, confirming that the core contributions of our work remain unhindered. The stress-testing against new baselines and ablation studies provided in the rebuttal *have only served to further solidify* the robustness and distinct novelty of LoVeC in efficient long-form confidence estimation.

Best regards,

The Authors

---

### Note · Authors · 2025-12-11

**Comment:**

In light of the recent OpenReview technical issue and the absence of reviewer replies to our rebuttal before the issue, we are withdrawing this submission to opt for other conferences.

**Withdrawal Confirmation:**

I have read and agree with the venue's withdrawal policy on behalf of myself and my co-authors.